# Flexibility and intrinsic disorder are conserved features of hepatitis C virus E2 glycoprotein

**Lenka Stejskal**[1,2], **William D. Lees**[2], **David S. Moss**[2], **Machaela Palor**[1], **Richard J. Bingham**[3], **Adrian J. Shepherd**[2]*, **Joe Grove**[1]*

**1** Institute of Immunity and Transplantation, Division of Infection and Immunity, University College London, London, United Kingdom, **2** Institute of Structural and Molecular Biology, Birkbeck College, London, United Kingdom, **3** Department of Biological Sciences, University of Huddersfield, Huddersfield, United Kingdom

* a.shepherd@mail.cryst.bbk.ac.uk (AJS); j.grove@ucl.ac.uk (JG)

**Data Availability Statement:** We have provided the results of our data analysis (which was used to generate the figures in the manuscript) as a supplementary file. The entire MD trajectory

## Abstract

The glycoproteins of hepatitis C virus, E1E2, are unlike any other viral fusion machinery yet described, and are the current focus of immunogen design in HCV vaccine development; thus, making E1E2 both scientifically and medically important. We used pre-existing, but fragmentary, structures to model a complete ectodomain of the major glycoprotein E2 from three strains of HCV. We then performed molecular dynamic simulations to explore the conformational landscape of E2, revealing a number of important features. Despite high sequence divergence, and subtle differences in the models, E2 from different strains behave similarly, possessing a stable core flanked by highly flexible regions, some of which perform essential functions such as receptor binding. Comparison with sequence data suggest that this consistent behaviour is conferred by a network of conserved residues that act as hinge and anchor points throughout E2. The variable regions (HVR-1, HVR-2 and VR-3) exhibit particularly high flexibility, and bioinformatic analysis suggests that HVR-1 is a putative intrinsically disordered protein region. Dynamic cross-correlation analyses demonstrate intramolecular communication and suggest that specific regions, such as HVR-1, can exert influence throughout E2. To support our computational approach we performed small-angle X-ray scattering with purified E2 ectodomain; this data was consistent with our MD experiments, suggesting a compact globular core with peripheral flexible regions. This work captures the dynamic behaviour of E2 and has direct relevance to the interaction of HCV with cell-surface receptors and neutralising antibodies.

## Author summary

Hepatitis C virus (HCV) is a globally important pathogen for which no vaccine is available. E2 is a protein found on the surface of HCV particles; it mediates interaction of HCV with cells and is a target for the human immune response. Current evidence suggests that antibodies targeting E2 are able to clear HCV infection, therefore, E2 is being pursued as a candidate vaccine. In this study we have built structural models of E2 from different strains of HCV and performed computational simulation to investigate how the E2 molecule moves. We have discovered that E2 possesses highly mobile regions; we propose that

datasets are available at https://zenodo.org/record/
3364033#.XV-qQ5NKhBx

**Funding:** LS received grant 109162/Z/15/Z from
the Wellcome Trust (https://wellcome.ac.uk). JG
received grant 107653/Z/15/Z from the Wellcome
Trust (https://wellcome.ac.uk) and Royal Society
(https://royalsociety.org). The funders had no role
in study design, data collection and analysis,
decision to publish, or preparation of the
manuscript.

**Competing interests:** The authors have declared
that no competing interests exist.

flexibility and disorder are defining characteristics of E2. This work provides a new perspective on E2 and will guide future studies into its basic functions and interactions with the immune system. Ultimately, our goal is to use this information to design new vaccine candidates by, for instance, locking the flexible regions of E2 such that they can be better targeted by antibodies.

## Introduction

The development of curative direct-acting antivirals has raised the possibility of eliminating hepatitis C virus (HCV) as a threat to global health and the current WHO target is a 90% reduction in chronic infections by 2030. However, as of 2018, 71 million people remain chronically infected, >80% of whom are unaware of their status, and transmission continues unabated with ~2 million new HCV infections every year. A vaccine capable of preventing chronic infection would significantly expedite current elimination programmes; consequently, there is renewed interest in determining the immune-correlates of viral clearance and developing immunogens capable of eliciting such responses [1]. These efforts have accumulated evidence that neutralising antibody responses targeting the major HCV glycoprotein, E2, protect individuals from chronic infection, thus placing E2 at the forefront of current HCV vaccine development [2–7].

E2 is presented on the surface of HCV virions as a heterodimer with a partner glycoprotein E1. These E1E2 dimers form higher-order oligomers, likely to be a trimer [8,9], which drive the processes of attachment, receptor engagement and fusion, to achieve virus entry. The molecular mechanisms by which E1E2 performs entry remain poorly understood, however, current evidence suggests that E2 is responsible for receptor engagement, whereas E1 is likely to contain the fusogen [8,10,11]. Whilst the structure of the E1E2 dimer has yet to be determined, crystallographic structures are available for the majority of the E2 ectodomain and the N-terminal portion of E1 [2,12–15]. These partial structures are sufficient to suggest that E1E2 possesses a unique protein fold and share limited structural homology to other viral fusion proteins. Consequently, E1E2 is unlikely to be a class I, II or III fusion machine and may represent a novel class of fusion machinery, although this has yet to be determined.

A wide variety of host factors have been implicated in HCV entry, however, the most well-supported and coherent model of HCV entry involves five essential host components [16–21]: CD81, scavenger receptor B1 (SR-B1), epidermal growth factor receptor (EGFR), claudin-1 (CLDN1) and occludin (OCLN). In this scheme, HCV particles first attach to the hepatocyte surface, via low-specificity interactions, and then assemble a receptor complex containing SR-B1, CD81 and EGFR; this complex undergoes actin-dependent translocation to the tight junction where HCV encounters CLDN1 and OCLN; from here HCV is internalised via clathrin-mediated endocytosis followed by pH-dependent fusion from the early endosome.

Of the five components necessary for entry, evidence suggests that only two are true receptors; direct E2 interaction being demonstrated for only SR-B1 and CD81 [17,21]. The majority of autologous and broad neutralising antibody (nAb) responses target the interactions between E2 and SR-B1/CD81. Moreover, only the initial stage of HCV entry (attachment and receptor engagement) are likely to be exposed to circulating antibodies; the latter events occurring in the cloistered environment of the tight junction [16]. Therefore, targeting these early receptor interactions with an immunogen is the most likely route to an effective HCV B-cell vaccine.

Importantly, soluble E2 (sE2), truncated to remove its transmembrane domain and expressed in the absence of E1, recapitulates HCV interactions with SR-B1, CD81 and various

antibodies; this has allowed functional, structural and biophysical characterisation of E2. For instance, alanine scanning mutagenesis, domain deletion and antibody blocking studies have identified the regions of E2 that contribute to receptor engagement. SR-B1 binding is likely to occur via hyper-variable region-1 (HVR-1), which is found at the N-terminal tail of E2 [17,22–25], whereas CD81 binding is thought to involve discontinuous regions that are brought together in the three-dimensional structure of E2 [12,26,27] (discussed in detail, below). Notably, there is evidence of an interdependence in E2-SR-B1/CD81 interactions, whereby SR-B1 binding may enhance E2-CD81 engagement [28,29], although the mechanism of this is unknown. Co-crystallization of sE2, or fragments thereof, with various antibodies has demonstrated that the most potent broadly nAbs target elements of the E2-CD81 interface [2,12,30–32]. Moreover, this structural information combined with biophysical measurements, such as hydrogen-deuterium exchange, suggest that E2 exhibits a high-degree of conformational plasticity and that this may present a barrier to the development of beneficial antibody specificities [33]. A greater appreciation of E2 dynamics is likely to inform our understanding of HCV-receptor interactions and their targeting by nAbs.

In this study we have used molecular dynamic (MD) simulation to explore the conformational landscape of the E2 ectodomain. Starting with partial crystal structures of E2 we modelled complete structures of the E2 ectodomain for three diverse strains of HCV: H77, 1b09 and J6. We then performed five independent 1μs MD simulations for each model; this revealed that individual E2 regions have distinct dynamics—some being very stable, whilst others exhibit high flexibility. This behaviour was very consistent between models, irrespective of their high sequence divergence, suggesting that the overall character of the protein is maintained between viral strains. Comparison of E2 flexibility and sequence conservation suggests that this flexible characteristic is imparted by highly conserved residues that act as pivot and anchor points to articulate the protein. Notably, the hypervariable regions, and in particular the N-terminal HVR-1, display very high flexibility and bioinformatic analyses indicate that HVR-1 has the hallmarks of an intrinsically disordered protein tail. Dynamic cross-correlation analysis revealed intramolecular communication: demonstrating concerted motions of different regions and suggesting that motion of the HVR-1 influences distant regions of E2. Finally, we used small-angle X-ray scattering to assess the folding state of soluble E2; these data were consistent with our MD experiments.

It is important to note that throughout this article we use the standard HCV numbering convention when describing E2: residues are numbered from the start of the HCV polyprotein and in reference to the prototypical H77 strain. Under this system, E2, which is 363 amino acids in length, is numbered 384–746. The E2 ectodomain, which is the focus of this study, is defined as residues 384–645.

## Results

### Generation of E2 ectodomain models

Over the past five years E2-antibody co-crystallisation approaches have yielded a variety of structures of varying completeness; the first being E2 from two prototypical strains, H77 and J6, solved by Kong, Wilson, Law et. al. [12,33], and Khan, Marcotrigiano et. al. [14], respectively (Fig 1 & S1 Fig). These structures were in good agreement and, together, reveal the basic organisation of E2. A central, IgG-like, β-Sandwich serves as a scaffold for the presentation of functionally important regions: i) an extended N-terminal stretch, termed the Front Layer, which is thought to contribute to CD81 binding; ii) a putative CD81 Binding Loop, which is presented in juxtaposition to the Front Layer by the two halves of the β-Sandwich; and iii) variable region 3 (VR-3, also known as the intergenotypic variable region) which also loops away

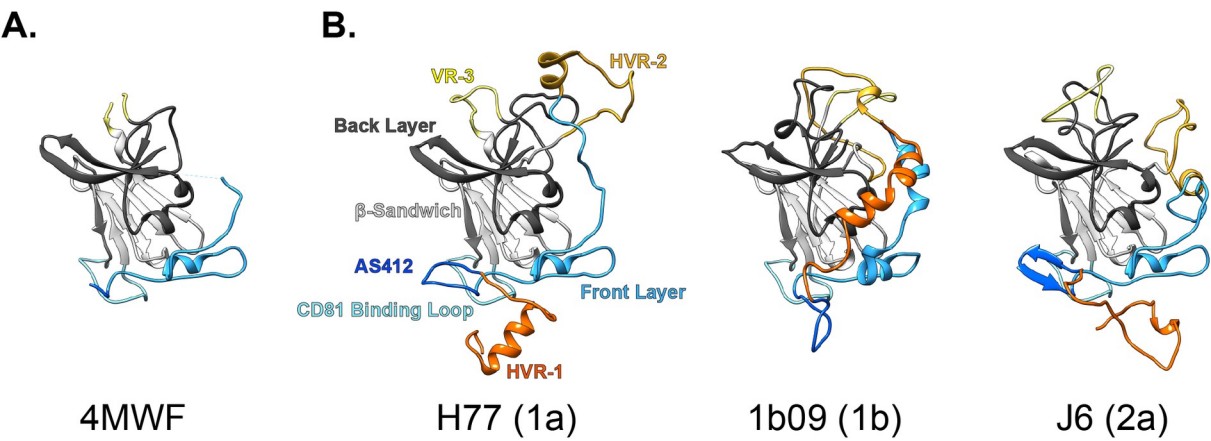

**Fig 1. E2 ectodomain models.** Partial crystal structures were used as the basis for building full length models of the E2 ectodomain (defined here as residues 384–645 of the HCV polyprotein). **A.** PDB 4MWF partial structure of H77 E2. **B.** Complete E2 ectodomain models of H77, 1b09 and J6 strains; annotations denote color-coding of regions, HCV genotypes are stated in parentheses.

from the central fold. The C-terminal part of the protein forms the Back Layer, which also has a high β-sheet content. Various disulphide bridges serve to cross-link the β-Sandwich and bridge the flanking regions (this is discussed in greater detail, below). Comparison of these initial structures with the RCSB Protein Data Bank (PDB) revealed little homology with known protein folds, suggesting that E2 (and, by extension, E1E2) is unlike any other viral fusion machinery yet described.

Whilst these structures encompass some important targets of nAbs (e.g. the Front Layer and CD81 Binding Loop), they also contained significant gaps and omissions arising from the protein design and/or poor electron density in the X-ray diffraction data. As a result these structures contain a discontinuous polypeptide chain and are poorly suited to MD simulations. Therefore, we sought to model the missing elements of the protein to create a complete structure of the E2 ectodomain. The first round of modelling drew upon all pre-existing structural information of E2, including structures of short E2 peptide epitopes from Antigenic Site 412 (AS412) in complex with various monoclonal antibodies (mAbs) [31,32,34–38]. The second round of modelling generated the missing N-terminal tail (i.e. HVR-1) and internal loops (e.g. HVR-2); this was performed in Rosetta using the Kinematic Closure (KIC) with fragments and Floppy Tail. The resultant complete models of H77 and J6 E2 are shown in Fig 1 and S1 Fig. During the course of our investigation Flyak, Bailey, Bjorkman et. al. solved a much more complete structure of the E2 ectodomain for the 1b09 strain of HCV (alongside a similarly complete structure of 1a53 strain E2) [2]. This only lacked the very N-terminal tail of E2, therefore, we modelled this missing region using Rosetta's Floppy Tail function (Fig 1 and S1 Fig), allowing 1b09 to be included in our study. Detailed information on the modelling strategy are provided in the Materials and Methods, the final models (as .pdb) are also provided (S1, S2 and S3 Files).

The E2 ectodomain models (Fig 1 and S1 Fig) exhibit broad agreement in their organisation: regions that contribute to CD81 binding (AS412, Front Layer and CD81 Binding Loop, colored-coded in shades of blue) are presented in close juxtaposition to create a putative CD81 binding interface. HVR-1 (orange) is in close apposition to this surface, whereas HVR-2 and VR-3 (yellows) loop away from an opposing face of E2. These flanking regions are disulphide cross-linked at various sites to the globular core of E2 (S2 Fig), which is comprised of the β-Sandwich and Back Layer. Whilst the overall arrangement is similar between models, there are

significant differences; S2 Fig provides comparison of secondary structure assignment, disulphide bridging and three-dimensional root-mean-square deviation of the polypeptide backbone (RMSD), which reflects the distance between corresponding residues in each model and serves as a measure of agreement. Further pairwise RMSD analysis between the E2 models and their parent crystal structures, and with E2 from alternative strains (1a53 and HK6a), are provided in S1 Table. There is good concordance in secondary structure and disulphide bonding (albeit with some subtle differences in cysteine cross-linking around VR-3), however, these characteristics are largely defined by the original crystal structures. The variable regions have the highest content of modelled residues; whilst being broadly similar in their secondary structure, these regions have the highest RMSD values (i.e. disagreement between models), due to differences in their arrangement. However, as discussed below, these regions are likely to be highly flexible and therefore it may be inappropriate to over-interpret the static conformations shown in our models. AS412 and the Front Layer exhibit intermediate RMSD values, but again various studies have demonstrated conformational heterogeneity in these regions [33,39]. Nonetheless, despite these subtle differences in the starting structures each strain behaved consistently upon MD simulation, as discussed below.

## Molecular dynamic simulations

E2 possesses 11 conserved potential N-glycosylation sites (PNGS), and analysis of sE2 exogenously expressed in mammalian cells suggests that the majority of these are occupied by complex and high-mannose glycans [40]. Consequently, glycans contribute a significant proportion of the total molecular mass of E2; however, due to the non-linear relationship between the number of atoms in a model and MD computation time, simulation of glycosylated E2 is likely to be computationally expensive [41]. Therefore, to explore the impact of glycosylation on E2 dynamics we generated a fully glycosylated model of J6 sE2 (Fig 2B and S3 Fig) for comparison with aglycosylated E2 in pilot MD simulations. We performed 5 independent 100ns explicit solvent MD simulations in Amber 16 using the GPU-based simulation engine (further details provided in the Materials and Methods). We used root-mean-square fluctuation (RMSF) as a summary statistic for comparison of the dynamic behaviour of both models; RMSF provides a per-residue measurement of distance moved relative to an arbitrary reference point (in this case, the average position of the given residue throughout an individual simulation). The RMSF profiles for aglycosylated and glycosylated J6 E2 models are remarkably similar (Fig 2A), with the majority of the protein displaying no significant difference in dynamics. Some regions, for instance the Front Layer, display reduced flexibility in the presence of glycans, however, the overall character of the protein remained unchanged. This is most apparent upon pairwise per-residue comparison (Fig 2C), which exhibits a very high level of correlation. Given these data, we opted to perform subsequent simulations with aglycosylated E2 models, allowing us to conduct multiple extended runs for H77, 1b09 and J6.

We performed five independent 1μs explicit solvent MD simulations in Amber 16 using the GPU-based simulation engine. Two representative trajectories for each model are provided in S1–S6 Movies. Fig 3A displays the mean per-residue RMSF values for each model, color coded by protein region; the resulting plots are very similar to those observed in the 100ns pilot runs (Fig 2A). The RMSF profiles reveal sharp transitions in mobility reflecting the distinct dynamics of different regions of E2 (Fig 3B); for instance, the β-Sandwich was very stable (average RMSF ~0.8Å), the CD81 Binding Loop had a small degree of flexibility (RMSF ~1.9Å), whereas HVR-1 was highly mobile (RMSF ~6Å). Notably, despite the relatively low sequence identity between H77, 1b09 and J6 (20–30% amino acid divergence, S4 Fig) and varying extents of modelled regions (S2 Fig), E2 exhibited consistent dynamic behaviour. This is most

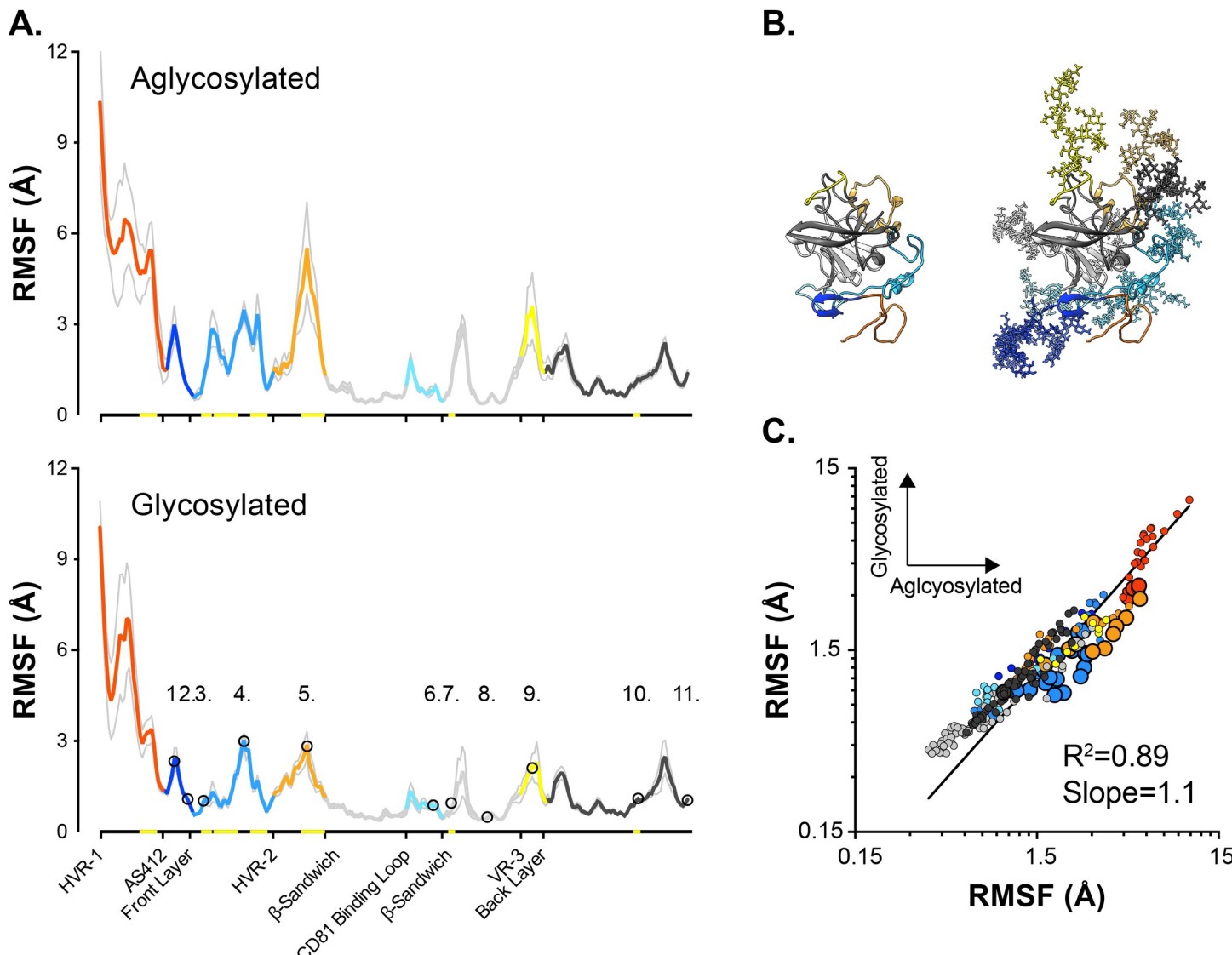

**Fig 2. Glycosylation has a modest effect on E2 dynamics.** Pilot 100ns MD simulations were performed with aglycosylated and glycosylated models of J6 E2. Five independent simulations were performed for each model. **A.** Average RMSF profiles (providing a measure of flexibility) for aglycosylated and glycosylated E2. Profiles are color-coded by region; grey lines indicate standard error of the mean. The 11 glycosylation sites are annotated as numbered circles. Yellow highlighting on x-axis denote significant difference between aglycosylated and glycosylated E2 (residue-by-residue comparison using an paired T-test, p <0.05, GraphPad Prism). **B.** Aglycosylated and glycosylated J6 E2 model structures. **C.** Pairwise residue-by-residue comparison of average RMSF values, larger data points indicate significant difference between aglycosylated and glycosylated E2. Data were fitted by log-log regression (GraphPad Prism).

apparent in Fig 3C, which provides pairwise comparison of RMSF values between strains; regression analysis indicating very good agreement ($R^2$ = 0.5–0.8, slope ~1).

## Conserved residues act as hinge and anchor points

The high consistency in behaviour between strains suggests that the dynamic characteristics of E2 are conserved despite high sequence variation. To explore this further we compared the distribution of highly conserved residues in relation to the RMSF profile of E2. To achieve this we first evaluated E2 sequence conservation by generating consensus protein sequences for the E2 ectodomain of HCV genotypes 1–6 (we omitted genotypes 7 and 8 due to the low numbers of

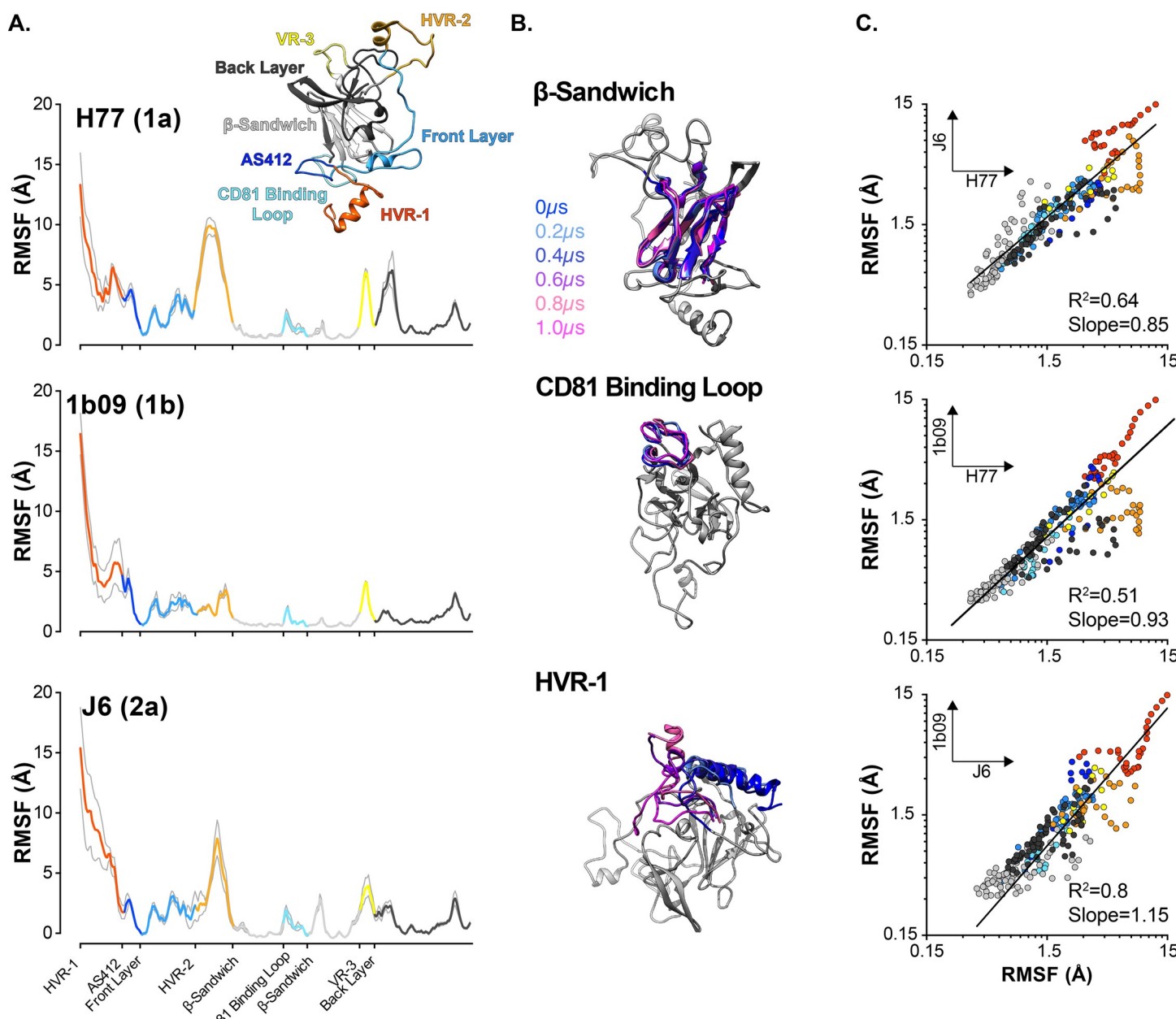

**Fig 3. E2 models exhibit consistent protein dynamics.** 1μs MD simulations were performed with aglycosylated H77, 1b09 and J6 E2 models. Five independent simulations were performed for each model. **A.** Average RMSF profiles for each model, color coded by protein region; grey lines indicate standard error of the mean. **B.** 200ns snapshots taken from a single representative H77 E2 trajectory. In each image the named protein region is color-coded by time; to provide context, the remainder of E2 is shown in grey for the T = 0μs snapshot alone. In each case, structures were aligned using the β-Sandwich as reference and were rotated to highlight the particular protein region. **C.** Pairwise residue-by-residue comparison of average RMSF values for each model. Data were fitted by log-log regression (GraphPad Prism).

sequences available [42]). These were then aligned to identify highly conserved residues (S5 Fig): 59.6% of E2 ectodomain residues are conserved across genotypes 1–6 E2 consensus sequences. As expected the different regions exhibited distinct sequence conservation profiles, however, even the variable regions contain highly conserved residues, suggesting functional constraints. We reasoned that transitions in protein dynamics along the polypeptide chain were likely to be mediated by particular amino acids: glycine bears a single hydrogen atom in

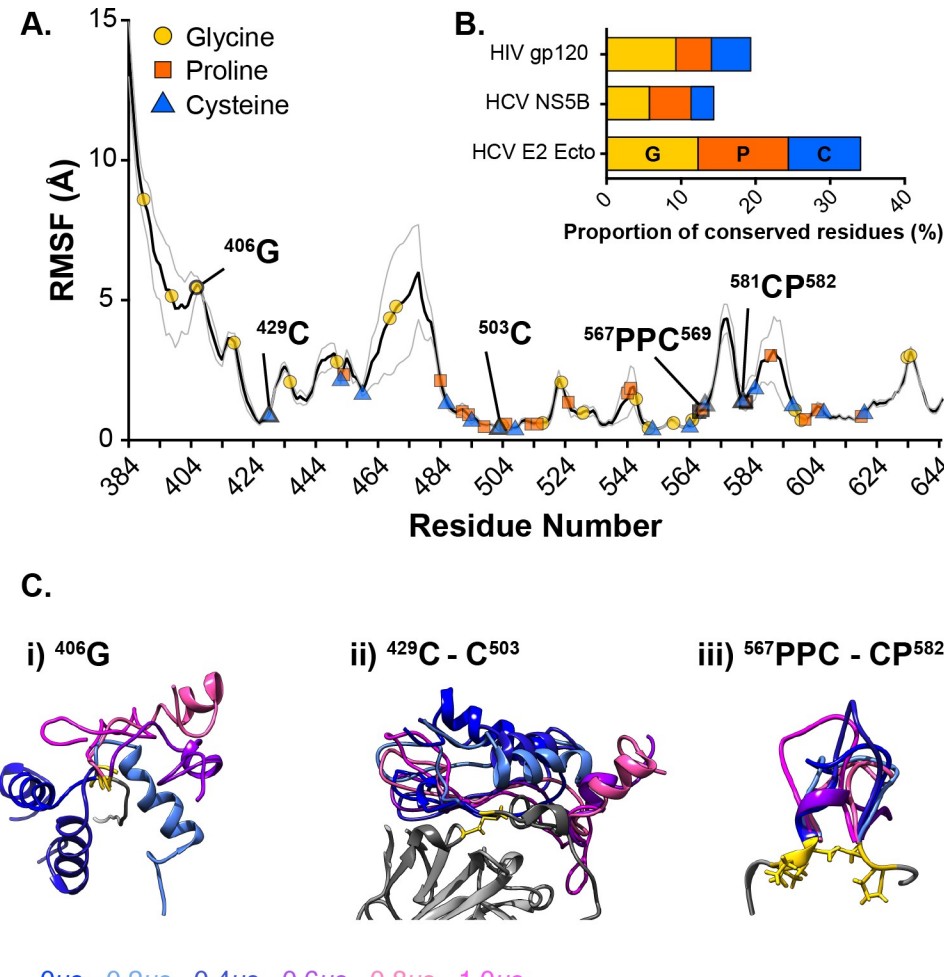

**Fig 4. Highly conserved residues act as hinges and tethers to articulate E2.** Comparison of E2 sequence data and RMSF profile reveal highly conserved residues that sit at transitions in protein flexibility. **A.** Average E2 RMSF—calculated by combining data from H77, 1b09 and J6—annotated with conserved glycines, prolines and cysteines. Specific example residues are highlighted. The grey line indicates standard error of the mean. **B.** The proportion of conserved residues that are either glycine, proline or cysteine in HIV gp120 (a viral glycoprotein), HCV NS5B (an RNA-dependent polymerase) or HCV E2 ectodomain. E2 exhibits an enrichment for these amino acids. **C.** Specific examples of hinge or tether points in E2 from a single representative H77 E2 trajectory, 200ns snapshots are shown. **i)** HVR-1 rotates around G406. **ii)** The C429-C503 disulphide bond anchors the N-terminus of E2 to the β-Sandwich. **iii)** Conserved proline and cysteine residues loop and isolate VR-3 from the core of E2. In each case, the region of interest is color-coded by time and the conserved residues highlighted in gold; to provide context, other selected elements of E2 are shown in grey for the T = 0μs snapshot alone. In each case, structures were aligned using reference residues surrounding the stated feature.

place of a side chain, this allows rotation around the Cα and therefore can impart high conformational flexibility; in contrast, the side chain of proline is bonded to both the Cα and amino group, this prevents rotation and therefore introduces rigidity; cysteines are capable of disulphide crosslinking, and therefore can tether protein regions to one another.

We overlaid the positions of highly conserved glycine, proline and cysteine residues along the average E2 RMSF profile (Fig 4A); this reveals the presence of conserved residues at certain transition points in the profile, suggesting that they are likely to act as hinges or anchors that facilitate the articulation of E2. Fig 4C provides some examples of these: i) G406 acts as a hinge point for rotation of HVR-1; ii) the disulphide bridge between C429 and C503 anchors the

flexible N-terminal portion of E2 (HVR-1 and AS412) to the stable β-Sandwich; iii) conserved prolines and cysteines sit either side of VR3 serving to loop it away from the main chain and isolate it from adjacent regions. These data suggest that the dynamic behaviour of E2 is maintained by a network of highly conserved residues. Indeed, glycines, prolines and cysteines are over-represented amongst the conserved residues when compared to an alternative HCV protein (NS5B) or an alternative viral glycoprotein (gp120) (Fig 4B); this suggests that E2 bears the evolutionary hallmarks of high protein flexibility.

### Conformational transitions in AS412

The crystal structure of AS412, a component of the CD81 binding site, has been solved multiple times, either in the context of the complete E2 ectodomain, or as an epitope peptide in complex with different mAbs. Taken together, these snapshot structures suggest a dynamic transition of AS412 from a closed β-hairpin conformation (e.g. PDB 4DGY, 6BZU, 4GAG) to an open extended conformation (e.g. PDB 4XVJ, 4WHY, 6MEJ) (Fig 5A) [2,31,32,34,35,39,43]. In the modelled structures of H77 and J6 E2, AS412 adopts a β-hairpin conformation, whereas in 1b09 AS412 forms a partially closed loop (derived from the original crystal structure, PDB 6MEI), which may represent an intermediate form between the closed and open states.

To explore whether conformational switching of AS412 is captured during MD simulation, we performed three-dimensional structural comparison (RMSD analysis) of AS412 from each MD trajectory to reference structures of β-hairpin (PDB 4DGY) and extended (PDB 4XVJ) conformers [34,35]. To visualise this data we plotted the two RMSD measurements for each frame against each other (50,000 frames per simulation); representative simulations are shown in Fig 5B, the data from all simulations are provided in S6 Fig. This analysis revealed a variety of behaviours. In some simulations AS412 was stable, for instance remaining as a β-hairpin (Fig 5Bi). However, in other simulations the closed conformation destablised and transitioned to an extended form, albeit with strain specific outcomes: in J6 and H77 AS412 tended to become more like the extended conformation identified in PDB 4XVJ (Fig 5Bii), whereas in 1b09 AS412 adopted a third conformation, which, although extended, was unlike PDB 4XVJ (Fig 5Biii).

To achieve an overview of the conformational landscape of AS412 we used the data from all simulations to generate RMSD heatmaps for each strain (Fig 5C and 5D); this reveals the occupancy of different conformational states. Here, J6 and H77 display consistent behaviour, with peak occupancy in the β-hairpin conformation, but with a tendency for this to transition to a PDB 4XVJ-like extended form, via a closed loop intermediate (similar to that observed in 6MEI). In 1b09, peak occupancy occurred in the closed loop conformation, with additional hot spots representing the alternative extended conformer.

Closer examination of the MD trajectories revealed further similarities between the simulated conformers and published structures of AS412. The extended conformation of AS412 captured in the PDB 6MEJ crystal structure (HCV strain 1a53 E2) is broadly similar to that seen in PDB 4XVJ with the notable exception of a C-terminal α-helix ($^{422}$L-R$^{424}$) [2]. We observed transient formation of the same α-helix, following destabilisation of the AS412 β-hairpin, in 3/5 simulations with J6 E2 (Fig 5E). Also, the extended crooked form of AS412 observed in 2/5 simulations with 1b09 E2 (Fig 5C and 5D) is similar to the extended form observed in crystal structure PDB 4WHY (Fig 5F) [32].

In summary, AS412 undergoes structural switching during MD simulation and explores conformational space that is consistent with the pre-existing crystal structures of this region. This is important for two reasons. First, the MD simulations are recapitulating experimentally

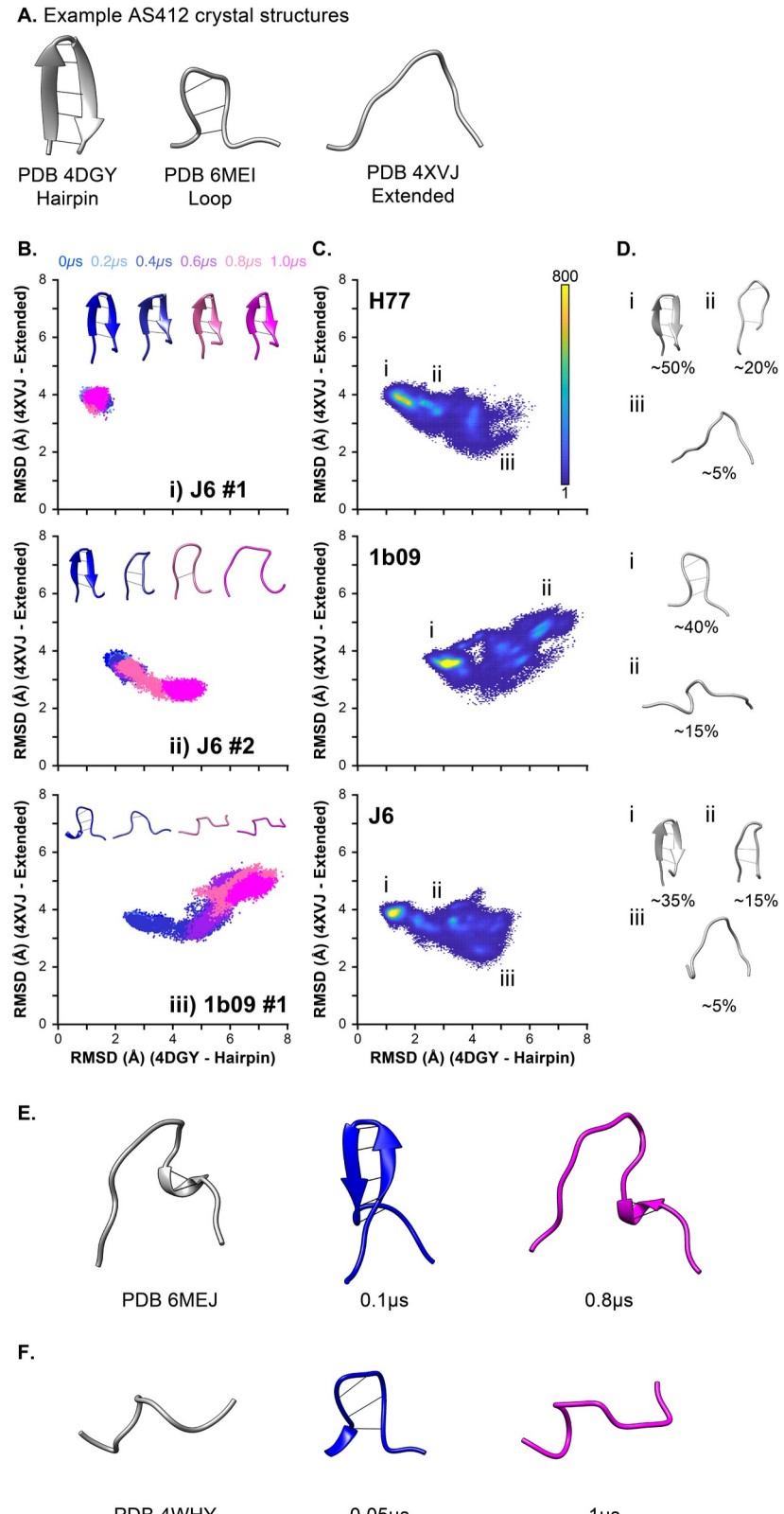

**Fig 5. Molecular dynamic simulations capture conformation transitions in AS412.** Crystal structures of E2 ectodomain, or E2 derived peptides, in complex with various antibodies have revealed conformational plasticity of

AS412. **A.** Representative structures of AS412 in a β-hairpin (PDB 4DGY), closed loop (PDB 6MEI) or extended (PDB 4XVJ) conformation. To assess conformational transitions of AS412 we measured backbone RMSD between each MD trajectory and reference structures in the β-hairpin (PDB 4DGY) or extended (PDB 4XVJ) conformations. **B.** Scatter plots of RMSD values for example simulations (as stated on the plot). The data points represent individual frames and are color-coded by time, as stated in the legend. Each plot includes representative AS412 structures from each simulation, also color-coded by time. **C.** RMSD scatter plot heat maps displaying the occupancy of AS412 in different conformational states across all simulations for each E2 model. The plots are color-coded for number of frames, as stated in the legend. **D.** Representative structures from different regions of the scatter plot (as annotated on C.), approximate values for the proportion of frames found within each state are provided. **E.** MD simulations recapitulate the extended AS412 conformation found in PDB 6MEJ, MD images are taken from a J6 E2 simulation. **F.** The extended conformation sampled in 1b09 simulations is similar to that observed in PDB 4WHY.

determined conformers of AS412; this provides validation of our experimental approach and suggests that MD is reliably simulating the behaviour of native E2. Secondly, these observations provide direct evidence that AS412 is dynamically sampling a range of defined conformations (as can be inferred from the various crystal structures); this is consistent with previous biophysical and MD analysis of AS412 peptides [44]. Therefore, the AS412 crystal structures solved in complex with different mAbs (e.g. PDB 4DGY—mAb HCV1, PDB 4XVJ–HC33.1) likely represent the capture of transient conformations of AS412 and are not due to induced-fit, in which mAb binding is forcing a non-native conformation. Given the importance of AS412 for CD81 binding, an obvious question is which of these conformations is active for CD81 engagement?

## Flexibility and disorder are intrinsic features of E2

HVR-1, HVR-2 and VR-3 represent peaks in the E2 RMSF plots (Figs 2, 3 & 4), suggesting a relationship between sequence diversity and protein flexibility. To investigate this further we compared RMSF to Shannon entropy, which provides a measure of sequence conservation, where high Shannon entropy indicates low conservation and vice versa [45]. Fig 6A displays the H77 E2 structure color coded for Shannon entropy (calculated from a multiple sequence alignment of GT1 E2 sequences); the variable regions are clearly visible as having high entropy. This pattern is mirrored in Fig 6B, which displays H77 E2 color coded for RMSF. To quantify this we examined the Shannon entropy of residues within flexible regions (arbitrary cut off of RMSF $\geq$3Å) or those within stable regions (RMSF <3Å); using this threshold, 15–30% of E2 residues are in regions of high flexibility. For each of our E2 models, residues in flexible regions exhibited significantly higher Shannon entropy, indicating an inverse correlation between flexibility and sequence conservation (Fig 6C). This may suggest that flexibility is an intrinsic feature of HCV variable regions.

Intrinsically disordered protein regions (IDPR) are important for the function and regulation of a wide variety of proteins, particularly in eukaryotes [45–48]. Given the high flexibility of the variable regions we asked whether E2 contains any putative IDPR. The sequences of each of our E2 models were bioinformatically analysed using the MetaDisorder MD2 server, which integrates the predictions of multiple independent algorithms to produce a consensus disorder tendency score for each residue [49]; this is shown in Fig 7. Here, residues with values >0.5 are likely to be intrinsically disordered; whilst the majority of E2 does not pass this threshold, HVR-1 possess a very high disorder tendency. Moreover, this is corroborated by analysis of the consensus E2 sequences for GT1-6. These bioinformatic predictions, and our MD data, suggest that HVR-1 is an intrinsically disordered protein tail; by analogy to IDPR in other protein systems, we may expect this feature of HVR-1 to be important for E2 functionality [50]. We should note that each of our models contained some helical secondary structure within HVR-1 (S2 Fig), which would argue against complete disorder; however, as is apparent

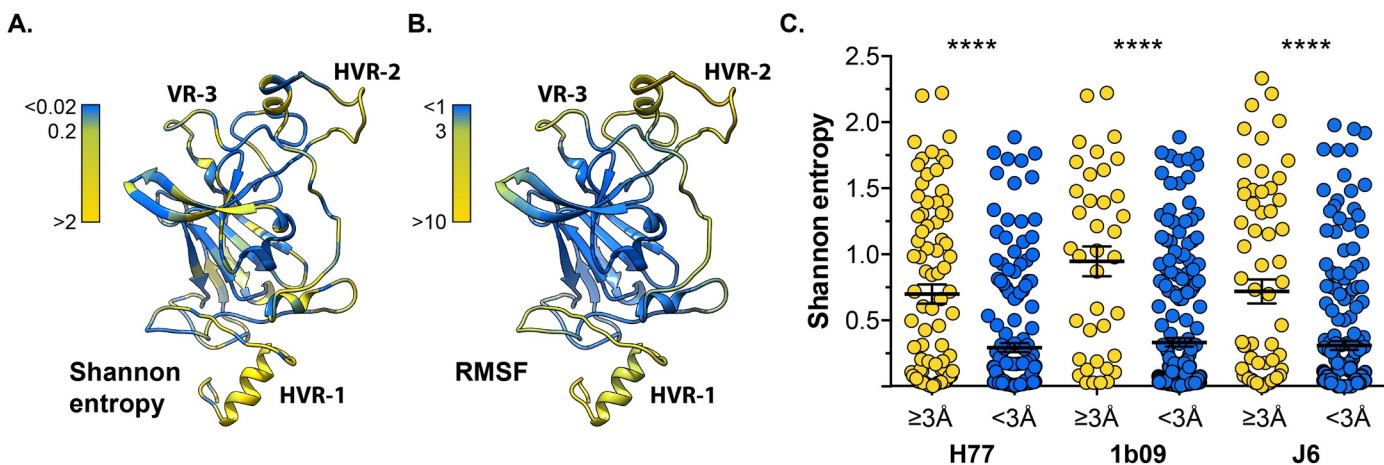

**Fig 6. Variable regions exhibit high flexibility.** Shannon entropy provides a measure of diversity with high values indicating low conservation. **A.** H77 E2 model color-coded for Shannon entropy; variable regions have high values. **B.** H77 E2 model color-coded for RMSF; variable regions exhibit high flexibility. **C.** E2 residues from H77, 1b09 and J6 were classified being in regions of high flexibility (RMSF $\geq$3Å) or low flexibility (RMSF <3Å). For each strain, residues within regions of high flexibility exhibited significantly higher Shannon entropy (Mann-Whitney test, p <0.0001, GraphPad Prism). Shannon entropy was calculated using E2 protein alignments for HCV genotype 1 (H77 and 1b09) and 2 (J6).

in S1–S6 Movies, this helical component frequently unravels giving rise to a random coil. This may suggest that HVR-1 can alternate between a flexible helix and complete disorder.

## Intramolecular communications allow E2 regions to influence each other

RMSF analysis provides a good measure of protein dynamics, however, it does not reveal intramolecular interactions. Therefore, we performed dynamic cross-correlation (DCC) analysis, which considers pairwise correlation of movement (i.e. the movement of each residue is compared with all others). Much like the RMSF plots, the DCC matrices for each strain are broadly similar (S7 Fig), therefore, we combined the individual plots into an average E2 correlation matrix (Fig 8A); examination of this reveals intramolecular communications throughout E2.

DCC matrices display the degree of correlation between each residue and every other residue in the protein, consequently they exhibit symmetry (Fig 8A). The diagonal yellow region in the DCC matrix indicates the high level of correlation along the polypeptide chain; this is a feature of all DCC matrices and demonstrates the coordinated motions of closely neighbouring residues. Other hotspots flank this central region; these represent secondary structure features that impart areas of local correlation. For example: i) α-helix $^{436}$G-F$^{442}$ in the Front Layer and ii) β-strand $^{551}$G-A$^{566}$ in the β-Sandwich appear as correlated regions that extend locally from the polypeptide chain (Fig 8B and 8C). Disulphide bridging links discontinuous regions of E2, resulting in distant correlations. For instance the residues surrounding iii) $^{429}$C-C$^{502}$ create a hotspot of high correlation, and iv) $^{459}$C-C$^{486}$ crosslinks the polypeptide chain either side of HVR-2, and therefore appear as discrete spots flanking this region (Fig 8B and 8C). Alongside these features, which one might expect given the structure of E2, our analyses reveal further interactions throughout E2. The Front Layer, for example, exhibits multiple peaks of correlation to the β-Sandwich and Back Layer suggesting concerted motions of these regions. HVR-2 and VR-3 exhibit comparatively little correlation with the rest of E2; this is consistent with the notion that movement of these regions is isolated from the core of E2 by disulphide crosslinking. This is in contrast to HVR-1, which exhibits negative correlations throughout E2 (Fig 8A, blue regions), indicating movement in the opposite direction. This would suggest that, unlike HVR-2 and VR-3, motions within HVR-1 have influence throughout E2.

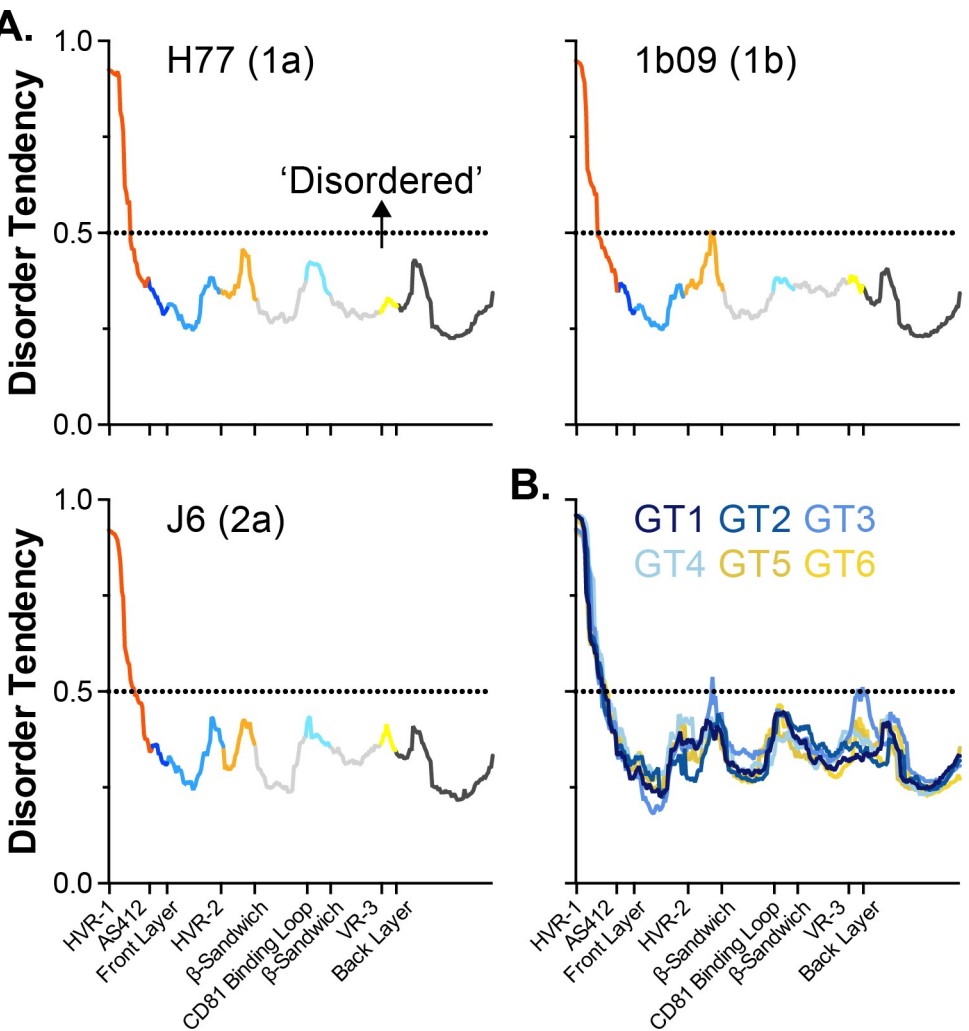

**Fig 7. HVR-1 is a putative intrinsically disordered protein region.** The MD2 server is a bioinformatic tool to assess protein sequences for their tendency to exhibit intrinsic disorder. Residues with MD2 scores above 0.5 are classified as disordered. **A.** MD2 scores for E2 sequences of H77, 1b09 and J6, profiles are color-coded by protein region. **B.** MD2 analysis of consensus E2 sequences from HCV genotypes 1–6. In all analyses, HVR-1 has a high disorder tendency.

## SAXS analyses of E2 are consistent with *in silico* experiments

Our computational and bioinformatics approach, in concert with various previous observations [39,51,52], provide good evidence that conformational plasticity and intrinsic disorder are defining features of the E2 glycoprotein. We sought to further verify this using biophysical analysis. Small-angle X-ray scattering (SAXS) is a low-resolution structural technique that, when performed on a solution of monodisperse protein, yields measurements of its size, shape and flexibility [53]. Using affinity purification and size exclusion chromatography, we produced high purity monodisperse J6 sE2 for SAXS analysis. Khan et al. had previously performed SAXS on a very similar J6 E2 ectodomain construct, this published data provides a point of comparison [14]. The radius of gyration ($R_g$) and maximum dimension ($D_{max}$) are measurements routinely extracted from SAXS data, both values are determined by the size of the particle being analysed. Khan et. al. reported an $R_g$ of 28.2Å and a $D_{max}$ of 84Å for J6 E2 ectodomain; our analyses yielded 30.87Å and 97Å respectively. Notably, Khan et. al. produced

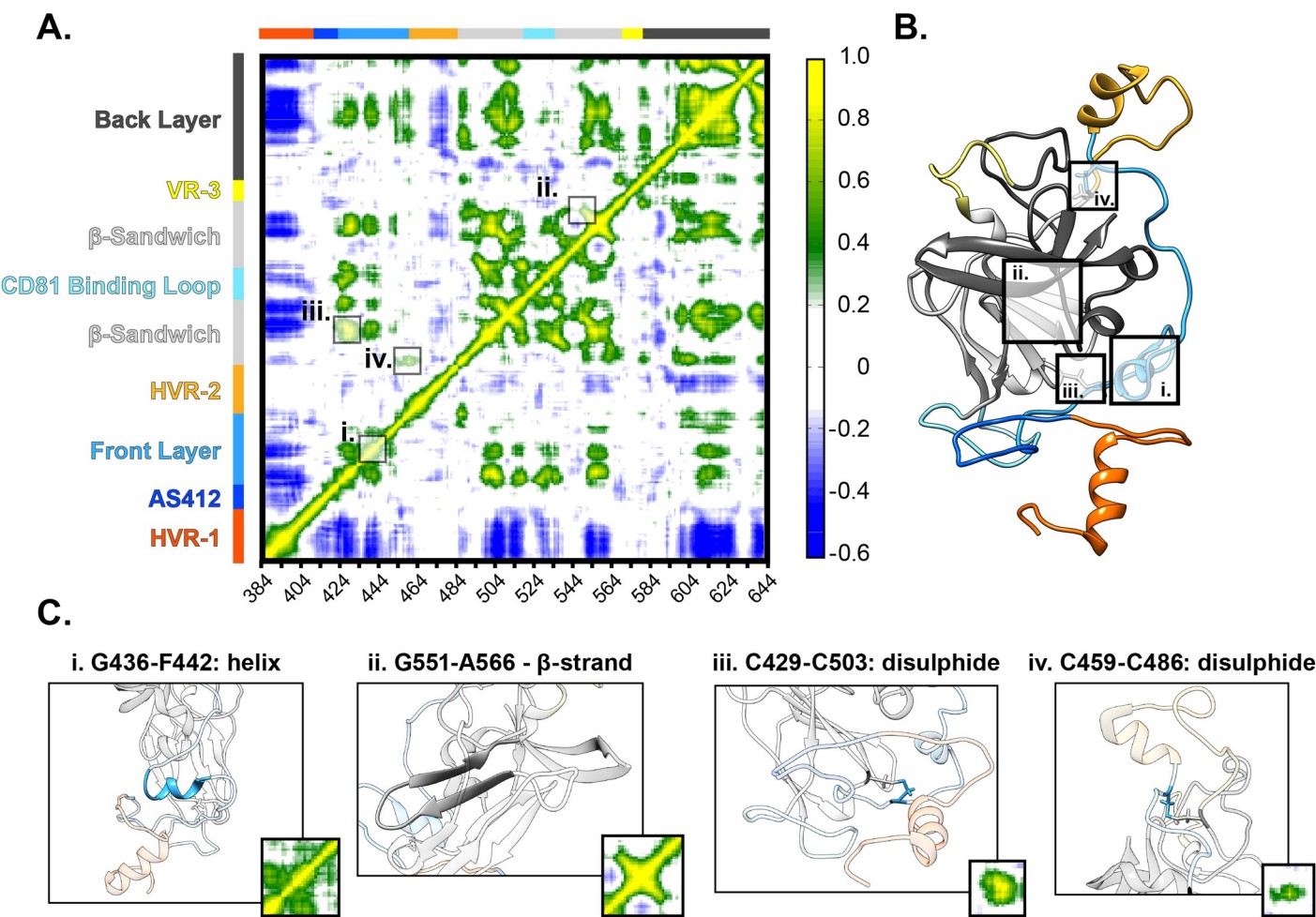

**Fig 8. Dynamic cross-correlation analysis reveals intramolecular communication.** DCC provides a residue-by-residue pairwise comparison of motion in MD trajectories to reveal correlations/anti-correlations in protein movement. **A.** An average DCC matrix for E2 –calculated by combining data from H77, 1b09 and J6. Color-coding indicates the degree of correlation. Specific features are highlighted. **B.** H77 E2 model annotated to highlight specific structural features. **C.** Images illustrating structural features and their corresponding DCC signatures. Helices and β-sheets (i and ii) impart local correlation, these appear diagonally relative to the central polypeptide chain. Disulphide bonds (iii and iv) can cross-link protein regions resulting in distant correlations.

J6 E2 in cells deficient for complex glycan synthesis (HEK293T GnTI-), consequently the mass of E2 was ~30% glycan; the expression system used in our study was proficient for complex glycan formation (HEK293T), resulting in a glycan component of ~50%. Given this, our data is in good agreement with that of Khan et. al.; the slightly larger values being likely attributable to the presence of complex glycans. Moreover, our experimentally determined $D_{max}$ value (97Å) is in excellent agreement with the largest atom-atom distance in our glycosylated model of J6 E2 (~100Å from glycan 2 to glycan 9, S3 Fig), further validating our data.

Next, the scattering data was evaluated for evidence of protein flexibility. Kratky plots display a mathematical transformation of particle scattering and encode information about the native protein fold; this approach is particularly good at distinguishing folded globular proteins from those that are largely unfolded, but can also detect intermediate forms with mixtures of order and disorder [54]. Kratky analysis of J6 sE2 (Fig 9) reveal a sharp peak at low angles, this is indicative of a globular fold and likely represents the central scaffold and closely associated regions (e.g β-Sandwich, Back Layer and Front Layer). For completely globular

proteins this peak is expected to decay to zero at higher angles. However, this is not the case for E2; at higher angles the data points level off and then slope upward, which is suggestive of a disordered component [55]. Therefore, SAXS analysis is largely consistent with our MD analysis, demonstrating that E2 is composed of a well-folded, globular core with evidence for peripheral regions that display flexibility and/or disorder.

## Discussion

There are numerous pieces of evidence that HCV E2 exhibits high conformational plasticity [39,51,52]. For example, initial E2 crystallization attempts required the removal of HVR-1, HVR-2 and AS412, this suggests they are not conducive to ordered crystal packing and, therefore, are likely to be flexible [12,14]. Moreover, even when present in the protein crystal these regions have not always resulted in defined electron density, again indicating disorder [2,14]. Structures of short fragments of AS412, or the Front Layer, in complex with various mAbs suggest alternative conformers for both regions [32,56]. Moreover, hydrogen-deuterium exchange, which provides insight to the solvent exposure and flexibility of a protein, has demonstrated that the Front Layer and CD81 Binding Loop are dynamic (HVR-1, AS412 and HVR-2 were missing from this analysis) [33]. These observations suggest plasticity, but provide only a fragmentary view of E2 behaviour. Whilst previous MD studies have yielded some understanding of E2 dynamism, experiments thus far have been limited to single short (100-500ns) simulations with incomplete protein structures from an individual strain of HCV [33,57]. In this study we have performed extensive repeat MD simulations of complete models from diverse HCV strains to gain a broad understanding of the conformational landscape of E2.

The first step in our investigation was to generate continuous models of the E2 ectodomain (i.e residues 384–645); this presented a challenge as the current structures of E2 are incomplete and/or fragmentary [2,12,14,39,44]. Therefore we opted for a hybrid modelling approach, first drawing upon all available E2 structures, then using appropriate functions in Rosetta to add the remaining regions. During the course of our investigation a more complete E2 structure was published by Flyak, Bailey, Bjorkman et. al.; this provided both another structure to include in our study and a point of comparison for our H77 and J6 E2 models [2]. The architecture of the three models are in good agreement (Fig 1 and S1 Fig), for instance, in their juxtaposition of the three elements of the CD81 binding site. Also, some modelled features are recapitulated in the more complete crystal structure, for example the C459-C486 disulphide bridge and the secondary structure assignment of HVR-2. Three structures of a GT6A (strain HK6a) E2 were also very recently published [13], however, these are very similar to the preexisting structures and provide no elements of E2 that have not already been solved; therefore, these 6A structures have limited use as a point of comparison. Nonetheless, superposition of PDB 6BKB (6A E2) with our models reveals RMSD values of 2.5–2.9Å (S1 Table), similar to those obtained using PDB 4MWF (original H77 E2 structure).

Despite the broad similarities between our models, there are some significant differences in the orientation of AS412, the arrangement of the C-terminal portion of the Front Layer and, consequently, in the presentation of HVR-2. We must also consider that all E2 structures have been solved by co-crystallization with mAbs; it is likely that this will also influence the apparent conformation of E2. Therefore, our findings come with the proviso that the E2 models are likely to include inaccuracies. Nonetheless, it is striking how consistent the models performed under MD simulation, as is apparent in both the RMSF profiles and DCC matrices. It is also worth noting that the J6 E2 structure (>50% modelled), behaved most similarly to the 1b09 E2

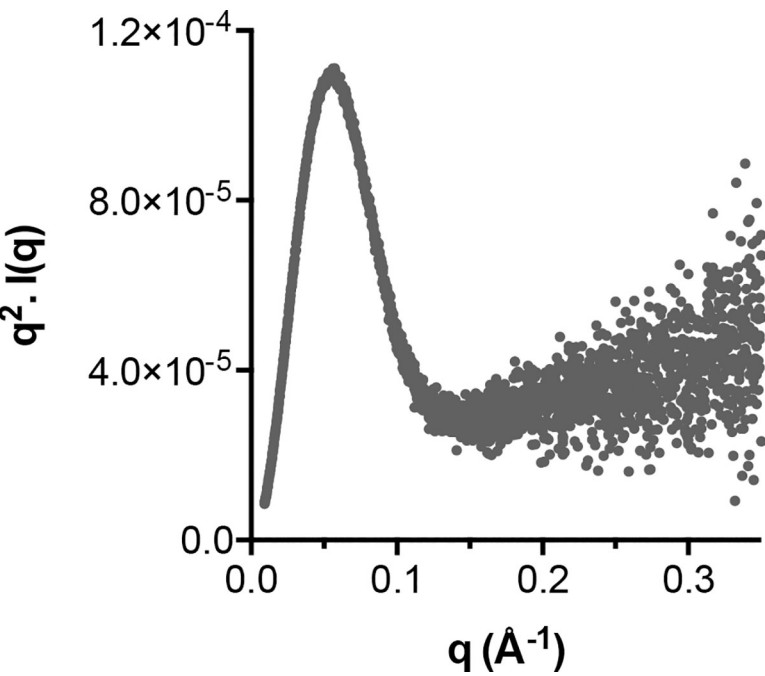

**Fig 9. Small-angle X-ray scattering by E2 is consistent with MD simulations.** SAXS is a low-resolution solution-based structural technique that can provide a measure of protein flexibility. Kratky analysis is an established approach to assess the state of protein folding. The Kratky plot of J6 soluble E2 is consistent with a globular core flanked by peripheral flexible regions.

structure (<10% modelled) (Fig 3C). This level of consistency between models suggests we have captured an informative view of E2 dynamism.

The different regions of E2 have been variously defined by sequence conservation (e.g. HVR-1), antigenicity (e.g. AS412), function (CD81 Binding Loop) and structure (e.g. β-Sandwich); despite these disparate definitions, our MD analysis suggests that each region has distinct dynamic characteristics. The N-terminal portion of E2, comprised of HVR-1 and AS412, represents a flexible tail that is anchored to the β-Sandwich by the disulphide bond between C429-C503. The Front Layer is disulphide bonded to both the β-Sandwich and Back Layer, however, this does not prevent it from exhibiting moderate flexibility; consistent with previous MD studies and hydrogen-deuterium exchange [33,57]. The CD81 Binding Loop, HVR-2, VR-3 are constrained only at their termini, where they loop away from the globular core of E2, consequently these regions are free to exhibit moderate to high flexibility. The β-Sandwich and, to some extent, the Back Layer act as a stable scaffold; this is particularly apparent in the DCC matrices, where these regions appear as centres of correlation that have points of influence throughout the rest of E2. Notably, in our pilot MD studies with glycosylated E2, glycans tended to reduce mobility of the underlying protein (Fig 2), likely through generating steric clashes, but this effect was not very strong. This may suggest that glycans exert only a modest influence on E2 protein dynamics, as has been reported for various other glycoproteins [58]. However, it is important to note that glycans are not inconsequential for E1E2 functionality; multiple studies have demonstrated a critical role for E2 glycans in HCV entry and immune evasion [59–62]. Moreover, a recent study suggested that glycans can modulate conformational shifts in E1E2 [63]. Therefore, it is likely that our experimental approach has not captured the functional importance of glycosylation.

The dynamic behaviour and modular architecture of E2 are maintained between the three strains of HCV included in our investigation, despite 20–30% sequence divergence. Analysis of sequence data suggest that these characteristics are imparted by a network of highly conserved residues that articulate E2. Our analysis focussed on glycine, proline and cysteine residues due to their potential to alter protein flexibility; indeed these amino acid types were often found at transition points in the E2 RMSF profile, indicating locations where they act as hinges or tethers (Fig 4). These amino acids are often conserved, due to their role in determining protein secondary and tertiary structure; nonetheless, comparison of their frequency in the E2 ectodomain with another HCV protein (NS5B) or another viral glycoprotein (gp120) would suggest that HCV E2 is enriched for conserved glycines, prolines and cysteines. This may suggest that E2 bears the evolutionary fingerprints of protein flexibility.

Experimentally determined structures of AS412 suggest that this region can adopt a range of defined conformations [39,44]. Comparison of our MD trajectories to these predetermined structures provided a means to validate our experimental approach and, conversely, the MD experiments allow us to directly assess conformational transitions in AS412. RMSD comparison of simulated E2 with structures of AS412 in the closed or open configurations revealed consistent conformational switching (Fig 5). The closed conformer was the most common form in each of our E2 models; for H77 and J6 this was a β-hairpin, for 1b09 this was a partially closed loop. However, in each model the closed form was relatively unstable, often unfolding into extended conformations. Importantly, the conformational space sampled during MD simulation was consistent with the published crystal structures, this would indicate that our experiments have faithfully reproduced the behaviour of native E2. Moreover, these observations suggest that the AS412 structures solved in complex with different neutralising mAbs represent snapshots captured from a dynamic equilibrium of different conformations. Indeed, the closed β-hairpin form of AS412 was the most commonly observed configuration during MD simulation, and is the most common form identified by mAb co-crystallisation suggesting an equilibrium shifted towards this closed form [39]. Therefore, in contrast to HVR-1, in which flexibility is driven by extreme disorder, the flexibility of AS412 is determined by conformational sampling, which is reflected in the relatively high RMSF of AS412 (Fig 3). Given its near complete conservation amongst genotypes (S5 Fig), it is probable that the programme of conformational transitions exhibited by AS412 is important for function, likely in the context CD81/nAb binding.

Dynamic cross-correlation analysis of MD trajectories provided a novel perspective on E2 by revealing intramolecular communications. Some correlations within E2 are completely intuitive, for example β-sheets result in concerted motions of the parallel peptide chains and appear as hotspots of correlation. However, the DCC matrices also reveal some features that are not easily predicted. For example disulphide bond C459-C486 appears as a discrete spot with few other correlations throughout E2 (Fig 8C iv), suggesting that its primary function is to isolate HVR-2. In contrast, the residues around C429-C503 have influence on regions throughout E2 (Fig 8C iii); this may provide an allosteric network by which motions are communicated across the protein. Furthermore, our DCC analysis indicated negative correlations associated with HVR-1, which to some extent mirror the hot spots associated with C429-C503. Taken together, this suggests that the N-terminal tail of E2, including the highly mobile HVR-1, exerts an influence throughout the rest of the protein, potentially via the disulphide tether at C429-C503.

As stated earlier, our primary motivation for studying E2 is to understand its molecular mechanics as both a viral entry machine and potential immunogen, therefore it is important to consider our findings in these contexts.

Some of the most potent and broadly cross-reactive neutralising mAbs target the CD81 binding site [2,12,13]; there is evidence such antibodies can drive the emergence of escape variants with reduced entry fitness resulting in viral clearance [6,64,65]. Therefore, these specificities are the current focus of B-cell vaccine design. CD81 binding is thought to require contributions from three elements of E2—AS412, the Front Layer and the CD81 Binding Loop —that are juxtaposed in to a putative CD81 interaction interface (Fig 1). However, it is notable that the CD81 binding site mAbs identified thus far are largely focussed on one or other of these elements; for example AR3C, HEPC3 and HEPC74 bind primarily to the Front Layer with only a minor contribution from the CD81 Binding Loop [2], whereas AP33, HCV-1 and HC33.1 are focussed entirely on AS412 [66]. This suggests that these elements are antigenically discontinuous. Our data indicate that each of these elements display intermediate flexibility and may sample multiple alternative conformations. Consequently, it is likely that the putative CD81 binding interface is somewhat incoherent and coalesces only upon receptor interaction (i.e. an induced-fit model of binding) or upon receiving other molecular triggers. This feature may guide immunofocus towards one or other of the individual elements of the CD81 binding site, but disfavour specificities targeting the entire interaction surface. Therefore, rational protein design to stabilise the CD81 binding site may result in a better immunogen, although a detailed structural understanding of E2-CD81 interaction may be necessary to guide such an approach.

An important prediction of our work is that HVR-1 is a highly flexible tail that bears the hallmarks of an intrinsically disordered protein region. IDPR play diverse roles in protein biology often mediating protein-protein interaction and/or modulating protein functionality [47]. For instance, a recent study demonstrated that the intrinsically disordered tail of UDP-α-D-glucose-6-dehydrogenase (UGDH) acts as an allosteric regulator of substrate affinity; in this example, structural constraint of the IDPR tail generates an entropic force which alters the dynamics and structure of UGDH [67]. HVR-1 may perform analogous functions for E2. Various studies demonstrate that removal of HVR-1 enhances HCV interaction with CD81 and mAbs; whilst this was originally believed to be due to removal of steric hindrances posed by HVR-1, it is becoming clear that this potential mechanism may be insufficient to explain the various data [63,68,69]. Although HCV-SR-B1 interactions may be complex in nature [25,70,71], HVR-1 is the putative E2-SR-B1 binding site [17,22–24], hence, we hypothesise that during virus entry HVR-1 transitions from a disordered and mobile state to one in which it is constrained in the E2-SR-B1 interface (S8 Fig). This transition may, in turn, alter the structure or dynamics of E2; indeed, the aforementioned DCC analysis provides evidence that motions within HVR-1 are communicated throughout the protein. This is a speculative hypothesis that requires targeted investigation, however, there is evidence that SR-B1 is able to enhance E2-CD81 interactions and that SR-B1-HVR-1 interactions result in priming of the HCV glycoproteins [28,29,63], to favour CD81 and/or mAb binding. Therefore, it is possible that E2-SR-B1 interaction provides a molecular trigger during HCV entry.

An important caveat to our work is that we are modelling the behaviour of monomeric E2 ectodomain, whereas on the surface of virus particles it is presented as an oligomer of E1E2 heterodimers. It is, therefore, very likely that certain regions of E2 are buried in the heterodimer and/or oligomer. Although there is no molecular structure for the E1E2 complex, various mutagenesis and evolutionary analyses has provided consistent reports on the E1-E2 interface, which is likely to include elements of HVR-2, the backlayer and the E2 stalk (the latter is not included in the ectodomain models) [9,72–75]. Therefore, although HVR-2 exhibits high flexibility in our simulations, it is likely to be constrained within the E1E2 complex. The only information available on the potential higher-order arrangement of E1E2 oligomers comes from a recent trimer model by Freedman, Houghton et. al., in which E1E2 dimers interact primarily

via the Back Layer, Stalk Region and Transmembrane Domain (the latter of which are not included in our ectodomain model) [8,9]. On the contrary, there are various pieces of evidence to indicate that the HVR-1, AS412, the Front Layer and CD81 Binding Loop are exposed and unconstrained on the surface of HCV particles, and we expect our MD analysis to be particularly relevant to understanding the behaviour of these functionally important regions. For example, these regions are freely available for antibody/receptor binding, and N-terminal genetically encoded tags attached to HVR-1 are readily accessible for particle purification and/or cleavage by proteases [76–78]. However, without a detailed structure of E1E2, we cannot be certain which regions are buried within the complex, or predict whether apparently flexible regions of E2 are stabilised in the context of the E1E2 oligomer.

To summarise, flexibility and disorder are defining features of the E2 glycoprotein. Its dynamic behaviour, captured in this study, may underpin its molecular functions as a viral entry machine and has important implications for the rational design of E2-based immunogens.

## Materials and methods

### Generation of complete E2 ectodomain models

Initial modelling was performed using the partial crystal structures of H77 E2 (PDB 4MWF, resolution 2.65Å) and J6 E2 (PDB 4WEB, resolution 2.4Å). 4MWF and 4WEB both lack the N-terminal portion of E2 and have various gaps in the polypeptide chain (S1 & S2 Figs). We generated complete models in a stepwise fashion.

**Structure refinement.** Prior to modelling we reassessed the pre-existing X-ray crystal structures; this was prompted by the fact that 4MWF contains an unusually high number of peptide bonds in the *cis* configuration (nine). Using Refmac and Coot (within the CCP4 suite) we rebuilt the H77 E2 structure with the *cis*-peptide bonds removed. For each of these refinements we assessed the R-factor as a measure of agreement between the experimental electron density and the refined model. Where removal of a *cis*-peptide improved agreement, the bond was kept in the *trans* configuration. Following these refinements only one *cis*-peptide remained (T512-P513) in 4MWF and 4WEB; notably this bond alone was found in the *cis* configuration in other recently published structures [2]. Analysis of the resultant refined structures demonstrated improved quality, apparent in the $R_{free}$ values: 4MWF = 0.274, 4MWF refined = 0.268, 4WEB = 0.268, 4WEB refined = 0.264.

**Disulphide assignments.** At the time of our initial modelling the most complete data on disulphide bonding came from 4MWF, therefore, this was used as a template for disulphide bonding in both our H77 and J6 model. The proximity of the regions surrounding C459 and C486 suggested bonding, therefore, an additional disulphide was created at this position, despite it not being resolved in 4MWF or 4WEB. Recently published E2 structures confirm a bond between C459 and C486. However, these structures also indicate bonding patterns around VR-3 that disagree with 4MWF (S2 Fig). This may represent natural heterogeneity in disulphide bonding or one of these configurations may be incorrect; nonetheless, this subtle difference appeared to have little impact on protein behaviour during MD simulations (Fig 3). Subsequent modelling steps were constrained to maintain disulphide bonding.

**Modelling.** Alongside the E2 core structures (4MWF and 4WEB) the structure of AS412 (residues 412–423) has been solved in complex with various monoclonal antibodies (PDBs 4DGV, 4DGY, 4G6A, 4GAG, 4GAJ, 4HS6, 4HS8, 4WHT, 4XVJ, 5EOC [31,32,34–38]). Therefore, these were also used as inputs for the generation of complete E2 ectodomain structures. We undertook an iterative process of refinement to generate our final models. During this process model quality was assessed by energy scoring and/or model clustering; this allowed the

selection of lead candidate models that proceeded to the next step. i) 1000 initial full length candidates were generated in Modeller v9.17 [79]; ii) Loopmodel class script was used to refine short modelled fragments (e.g. 574–577 and 586–596) resulting in a further 1000 candidates; iii) The lead candidate model was further processed in Rosetta v3.7 [80]: following relaxation of the structure, the HVR-2 loop region (453–491) was remodelled, using kinematic loop closure with fragments [81], to create 10,000 candidates. These were scored using the talaris2014 Rosetta scoring function and the 1000 best scoring models were clustered using Calibur [82], the centroid of the most populated cluster was selected for further processing. iv) Finally HVR-1 (384–412) was further modelled using Floppy Tail [83], resulting in 10,000 candidates. These were then selected by Calibur clustering, as described above, to produce a final full-length model. At a later stage of our study a more complete E2 structure was published for the 1b09 strain of HCV (PDB 6MEI, resolution 2.9Å). Therefore, to allow this to be included in our investigation we modelled the missing N-terminal residues (384–410) using the Floppy Tail function, as described above in step iv. Note that Flyak et. al. also published a largely complete structure for the 1a53 E2 ectodomain (PDB 6MEJ), however, this structure would have required modelling of both HVR-1 and VR-3, therefore we opted for the 1b09 structure (PDB 6MEI) as it was continuous and only required modelling of the N-terminal HVR-1. The final models for H77, 1b09 and J6 are provided as S1–S3 Files. The software scripts used to model E2 are also provided as S4 File.

Model analysis (e.g. secondary structure assignment and backbone RMSD) and generation of all structural images were performed using UCSF Chimera, developed by the Resource for Biocomputing, Visualization, and Informatics at the University of California, San Francisco, with support from NIH P41-GM103311 [84].

## Molecular dynamic simulations

We performed explicit solvent MD simulations in Amber 16 using the GPU-based simulation engine [85,86]. The models were solvated in a truncated octahedral box using OPC water molecules. The minimal distance between the models and the box boundary was set to 12 Å with box volume of $4.2 \times 10^5$ Å$^3$, $4.8 \times 10^5$ Å$^3$, $3.5 \times 10^5$ Å$^3$. Simulations were performed on GPUs using the CUDA version of PMEMD in Amber 16 with periodic boundary conditions. To retain disulphide bonds during the simulation, CONECT records were created using the MakeConnects.py script, which is part of the AmberUtils package (https://github.com/williamdlees/AmberUtils). MolProbity software was used to generate physiologically relevant protonation states.

**Minimisation and equilibration.** Protein models need to be put through a series of minimization and equilibration steps in order to be prepared for MD simulation. This removes any steric clashes and slowly heats the system to a physiological temperature.

The systems were minimised by 1000 steps of the steepest descent method followed by 9000 steps of the conjugate gradients method. Sequential 1ns relaxation steps were performed using the Lagevin thermostat to increase the temperature from 0 to 310K, with initial velocities being sampled from the Boltzmann distribution. Pressure was kept constant using the Berendsen barostat. During these steps atoms were restrained by a force of 100 kcal/mol/Å$^2$. The restraint force was decreased to 10kcal/mol/Å$^2$ during a subsequent 1ns equilibration at 310K.

A further minimisation step was included with 1000 steps of the steepest descent method followed by 9000 steps of the conjugate gradients method with atoms restrained by a force of 10 kcal/mol/Å$^2$. The systems were then subjected to four 1ns long equilibration steps at constant pressure with stepwise 10-fold reduction of restraint force from 10 to 0kcal/mol/Å$^2$. All minimisation and equilibration stages were performed with a 1fs time step.

**Production runs.**   For each model, initial 1μs production runs were simulated under constant volume and temperature using the Langevin thermostat, with a 4fs time step. Short-range cutoff distance for van der Waals interactions was set to be 10Å. The long-distance electrostatics were calculated using the particle mesh Ewald method. To avoid the overflow of coordinates, iwrap was set to 1. Default values were used for other modelling parameters. To achieve independent repeat simulations, we performed steps to decorrelate the output from the equilibration process. The coordinates, but not velocities, from the final equilibration step were used as input for a short (40ns) production run, with velocities being assigned from the Boltzmann distribution using a random seed. The coordinates, but not velocities, from this run were used for a second 4ns production run, with velocities assigned as above. The coordinates and velocities from this run were then used as input for a 1μs production run. This process was repeated for each independent simulation.

For MD of glycosylated E2, 11 N-linked glycans were modelled on to the relaxed and equilibrated J6 E2 model using the GLYCAM glycoprotein builder. The glycan species were as defined by Urbanowicz et. al. [40]. For these simulations, 100ns production runs were performed with a 2fs time step.

The MD trajectories were analysed using scripts available in cpptraj from Amber Tools 16. For RMSF analyses, the average structure generated from the given trajectory was used as the reference structure. The analyses were performed using the backbone Cα, C and N atoms.

Note that during the review of this work it was identified that two C-terminal amino acids (isoleucine and glycine) were erroneously included in the 1b09 model, having originated from the parent crystal structure PDB 6MEI; in native E2 these residues are conserved as tryptophan and threonine. However, it was decided to not rerun the MD simulations as the impact of these erroneous residues is likely to be negligible: scrutiny of our DCC analysis (Fig 8) suggests that these terminal residues exert little to no influence on the rest of the protein with mean correlation scores of 0.1 and 0.07 respectively.

## Sequence analysis

Pre-aligned protein sequences for HCV genotypes 1–6 E2 and NS5B were downloaded from the HCV GLUE database, alignments were then filtered to remove redundant sequences with a >98% identity threshold using Jalview 2 [87]. Consensus sequences for each protein were generated and aligned in Unipro UGENE, allowing the assessment of conservation. The same analysis was also applied using pre-existing consensus gp120 sequences representing all major clades of Group M HIV-1. Shannon entropy was determined for HCV genotypes 1 and 2 E2 by analysing the respective protein alignments using the Entropy-One tool on the HCV Los Alamos database.

## Soluble E2 production and SAXS

Soluble J6 E2 consists of residues 384–661 of the HCV genome flanked by an N-terminal tissue plasminogen activator signal sequence (which targets E2 for secretion, and is cleaved by signal peptidases) and a C-terminal Twin-Strep-tag (which allows affinity purification and detection). HEK 293T cells, CRISPR-Cas9 edited to prevent CD81 expression, were transduced with lentivirus encoding sE2. Cell culture media, containing sE2, were harvested every 24 hours for up to 6 weeks, samples being frozen immediately at -80˚C. High purity monomeric sE2 was generated by sequential affinity purification using StrepTactin-XT columns and size-exclusion chromatography. For quality control, purified sE2 was assessed for antibody reactivity by ELISA and receptor binding activity by flow cytometry.

SAXS experiments were performed at the B21 Synchrotron Radiation Beamline at the Diamond Light Source under the project SM22783-1. 50µl of 1mg/ml sE2 (in 50 mM Tris/HCl, 150 mM NaCl, 1% glycerol, pH 7.2) was passed through a 37˚C 1.5mm quartz capillary for exposure to the X-ray beam. The beam at the sample position was 1 mm × 2 mm and had an incident flux of $4 \times 1012$ photons/s at 13.1 keV ($\lambda = 0.9464$ Å). 10 frames (with 3s exposure time) were collected using the Dectris Eiger 4M detector at a distance of 4.014 m from the sample. The data were reduced from 2D to I(Q) versus Q (where Q = 4π(sin θ)/λ) using the in-house software DAWN 10.0.17. A blank buffer only sample was also acquired for background subtraction.

Scattering data was visually checked for aggregation, radiation damage, interparticle interference and consistency. The Guinier analysis, using the PRIMUS software included in the ATSAS package 2.8.3, was used to determine the reciprocal space forward scattering I(0) and Rg for each sample by defining a linear fit at low Q region (Q x Rg < 1.3). This indicated that there was no significant aggregation present in any of the samples. The dimensionless Kratky plot was generated using the Scatter software [88].

## Supporting information

**S1 Table. E2 structure RMSD comparison. A.** Pairwise backbone RMSD comparison of each E2 model with its parent crystal structure and with E2 from alternative strains (1a53 and HK6A). **B.** As above, but with the analysis limited to the β-sandwich; this provides a degree of agreement between an unmodelled portion of the protein.
(TIF)

**S1 Fig. E2 ectodomain models.** Partial crystal structures were used as the basis for building full length models of the E2 ectodomain **A.** Partial crystal structures of E2 from H77 (PDB 4MWF), 1b09 (PDB 6MEI) and J6 (PDB 4WEB). **B.** Corresponding complete E2 ectodomain models shown at 0˚, 90˚ and 180˚ rotation. Backbone RMSD values reflect divergence between crystal structures and models; all RMSD values are low, indicating good agreement. **C.** H77 E2 ectodomain sequence organised by protein region. Numbering (relative to start of HCV polyprotein) defines region assignments.
(TIF)

**S2 Fig. E2 structure comparison. A.** Aligned protein sequence from each model, organised by region, modelled regions are shaded in grey and conserved cysteine residues are orange. Secondary structure assignments are shown above the sequences. **B.** Disulphide bonding pattern for each model, cysteine positions are numbered according to the H77 reference sequence. **C.** H77 ectodomain model color coded to display backbone RMSD between models; region names are annotated with their average RMSD value (the mean RMSD of all residues within a given region). High RMSD values indicate disagreement between the models. For RMSD analysis, model structures were aligned using the β-Sandwich as a reference.
(TIF)

**S3 Fig. Glycosylated J6 E2.** Glycans are color coded according to protein region. Models are shown at 0˚ and 200˚ rotation.
(TIF)

**S4 Fig. E2 ectodomain sequence alignment.** Protein sequence is organised by region. Bars indicate level of conservation at each position. Tables indicate pairwise protein homology for the entire E2 ectodomain and without HVR-1, which is the major source of divergence. Values represent % identity and % similarities in parentheses (taking into account the equivalencies of

certain amino acids).
(TIF)

**S5 Fig. Genotypes 1–6 E2 ectodomain alignment.** Aligned consensus E2 sequences from HCV genotypes 1–6, organised by protein region. Bars indicate level of conservation at each position. The consensus sequence (Con) indicates residues that are conserved in all 6 sequences.
(TIF)

**S6 Fig. AS412 RMSD plots for each simulation.** Scatter plots of backbone RMSD values between each MD trajectory and reference structures in the β-hairpin (PDB 4DGY) or extended (PDB 4XVJ) conformations. The data points represent individual frames and are color-coded by time, as stated in the legend.
(TIF)

**S7 Fig. E2 DCC matrices.** DCC provides a residue-by-residue pairwise comparison of motion in MD trajectories to reveal correlations/anti-correlations in protein movement. DCC analysis was performed on MD data from H77, 1b09 and J6. Color-coding indicates the degree of correlation.
(TIF)

**S8 Fig. Hypothesis: HVR-1 may transition to a constrained state during virus entry.** SR-B1 is a receptor for HCV that interacts with E2 via HVR-1. Therefore, it is likely that the flexible and largely disordered HVR-1 will become constrained upon interaction. This may provide a mechanism by which receptor binding is communicated to the rest of E2. Image depicts H77 E2 with alternative conformations of HVR-1 (color coded by time, as in Fig 3) and a homology model of SR-B1 based on the structure of LIMP-2 (PDB 4F7B).
(TIF)

**S1 File. H77 E2 Model.** Final model of H77 E2 ectodomain used in this study.
(PDB)

**S2 File. 1b09 E2 Model.** Final model of 1b09 E2 ectodomain used in this study.
(PDB)

**S3 File. J6 E2 Model.** Final model of J6 E2 ectodomain used in this study.
(PDB)

**S4 File. Modelling scripts.** Modeller and Rosetta software scripts used to create E2 models.
(DOCX)

**S5 File. Underlying Data.** Results of data analysis presented in manuscipt figures.
(XLSX)

**S1 Movie. H77 A.** Movie of a representative 1μs H77 E2 MD simulation.
(MPG)

**S2 Movie. H77 B.** Movie of a representative 1μs H77 E2 MD simulation.
(MPG)

**S3 Movie. 1b09 A.** Movie of a representative 1μs 1b09 E2 MD simulation.
(MPG)

**S4 Movie. 1b09 B.** Movie of a representative 1μs 1b09 E2 MD simulation.
(MPG)

**S5 Movie. J6 A.** Movie of a representative 1μs J6 E2 MD simulation.
(MPG)

**S6 Movie. J6 B.** Movie of a representative 1μs J6 E2 MD simulation.
(MPG)

## Acknowledgments

We would like to thank Nathan Cowiesan for his support during the SAXS experiments; Richard Goldstein and Greg Towers for critical evaluation of the study; and Mphatso Kalemera and Tobias Starling for their contributions to our research.

## Author Contributions

**Conceptualization:** Adrian J. Shepherd, Joe Grove.

**Formal analysis:** Lenka Stejskal, Joe Grove.

**Funding acquisition:** Joe Grove.

**Investigation:** Lenka Stejskal, Machaela Palor, Richard J. Bingham.

**Methodology:** Lenka Stejskal, William D. Lees, David S. Moss.

**Project administration:** Joe Grove.

**Software:** William D. Lees.

**Supervision:** Adrian J. Shepherd, Joe Grove.

**Writing – original draft:** Joe Grove.

**Writing – review & editing:** Lenka Stejskal, Adrian J. Shepherd, Joe Grove.

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
