## [Decision Letter · Decision Letter 0]

16 Sep 2019

Dear Dr Grove,

Thank you very much for submitting your manuscript 'Flexibility and intrinsic disorder are conserved features of hepatitis C virus E2 glycoprotein.' for review by PLOS Computational Biology. Your manuscript has been fully evaluated by the PLOS Computational Biology editorial team and in this case also by independent peer reviewers. The reviewers appreciated the attention to an important problem, but raised some substantial concerns about the manuscript as it currently stands. While your manuscript cannot be accepted in its present form, we are willing to consider a revised version in which the issues raised by the reviewers have been adequately addressed. We cannot, of course, promise publication at that time.

Sincerely,

Guanghong Wei

Associate Editor

PLOS Computational Biology

Rob De Boer

Deputy Editor

PLOS Computational Biology

[LINK]

Reviewer's Responses to Questions

**Comments to the Authors:**

Reviewer #1: The authors provide an in-depth study of the dynamics of the HCV E2 ectodomain, relating it to key sequence features, as well as bnAb and coreceptor engagement. While this provides new insights into HCV E2 and its function, there are some issues which may limit the accuracy of the modeling, including lack of E1 glycoprotein in the models, which interacts with E2 in the native virion, likely at some E2 sites that are considered in this study. Given the lack of an experimentally determined E1-E2 co-crystal structure, this was a necessary omission, and the authors do note this shortcoming in the Discussion. Overall, this study is a useful addition to the field, but several comments should be addressed by the authors to improve the clarity and to support the conclusions of this study:

Major comments:

1. It is concerning that there does not seem to be a way readily to gauge the accuracy of the molecular dynamics simulations and the readout, including the RMSF information (Figs. 2 and 3), to ensure that the modeling represents true E2 dynamic behavior on the virion. The authors could, for instance, assess (based on backbone RMSD) whether sampled or preferred states of AS412 from MD simulations recapitulate the states observed in x-ray structures of that epitope engaged by bnAbs HCV1 and HC33.1, or alternatively they can assess against the x-ray structure of the AS434 epitope engaged by bnAb HC84.1. It is possible that lack of preference for these conformations in the MD simulations could point to “induced fit” behavior, rather than a shortcoming of the simulations themselves. At very least, the authors should make a brief note of this uncertainty, if they are unable to directly demonstrate through correspondence with experimental structural or other data that the MD simulations are accurate representations of native E2.

2. For the consensus sequence generation, it is not clear whether the authors performed any analysis or filtering to avoid redundancy or over-representation of certain sequences or subtypes, e.g. by culling E2 sequences with >98% identity to any others in the dataset. This would reduce the effect of any sampling bias in the HCV sequence databases. Such methods should be noted explicitly by the authors. If no redundancy filter or criterion was used, the authors should consider using one, or they should at very least note why one would not be needed.

3. To further investigate sequence diversity and structural/dynamic conservation, it would be reasonable to include one of the genotype 6 E2 core x-ray structures reported by Tzarum et al (reference 13), which were released in the Protein Data Bank late last year. It is possible that the relatively recent release of these structures prevented analysis by the authors for this study. However, the authors can mention and include basic analysis of one representative gt6 E2 structure, if time does not permit molecular dynamics simulations of such a structure to be included in this study.

4. The Modeling methods section requires more details for readers to understand and potentially to utilize these methods in their own work. For instance, what versions of Modeller and Rosetta were used? It is noted that models were scored “using talaris2014”, and most readers may not understand that this is a scoring function in Rosetta. For reproducibility and clarity, the authors should provide the details of the command line parameters that were used for Modeller, LoopModel, FloppyTail, etc., for instance as reported by Adolf-Bryfogle et al. PLoS Comp. Biology 2018 14(4):e1006112 in their supporting information text.

5. The 1b09 E2 ectodomain model PDB, provided as a supplemental file, seems to have erroneous residues near the C-term, possibly an artifact from the x-ray structure. The C-terminal residues should be “NWT”, but instead they are “NIT”. It is suggested that, if this error is confirmed, that simulations be re-run, or at very least the authors should state in the text why it is inconsequential.

Minor comments:

6. Line 130. It seems that E2 should be residues 384-746, not 384-747.

7. Line 162. 1a53 and 1b09 E2 ectodomain structures were reported by Flyak et al., not just 1a53. Evidently the authors had a typo where 1b09 was replaced by 1a53 throughout the text, which they plan to correct, but in any case, they should note both isolates used in the set of structures reported by Flyak et al.

8. Lines 186-187. “…but again various studies have demonstrated conformational heterogeneity in these regions”. Citations should be given here.

9. Line 198. “unglycosylated” is not commonly used, “aglycosylated” should probably be used here for clarity.

10. Lines 221-222. A “~30% amino acid divergence” is noted between the isolates studied here, and there is a reference to a multiple sequence alignment in Fig. S4. It would be helpful for readers to better understand the sequence divergence if the authors can include the actual pairwise sequence identities between E2 ectodomains for these three isolates (H77, J6, 1b09) in a table, optionally including the sequence % identities with and without HVR1.

11. Line 238. “in place of a side change” should be “in place of a side chain”.

12. Lines 356-357. “…generate full-length models of E2; … no known structural homologues of E2.” It seems that rather than “full-length” E2, the authors are only modeling E2 ectodomains, omitting the transmembrane regions, and also omitting the C-terminal “stem” regions of the ectodomains (aa 646-717). To avoid reader confusion, the authors should change this wording, and possibly clarify what is missing. Regarding the absence of “known structural homologues of E2”, there are several x-ray structures of E2 itself which the authors used directly to generate their models, as noted by the authors in the next sentence. The authors did need to model missing loops and the N-terminus, which can be considered a challenge, but having no known E2 structural homologues should not be stated as a challenge from a modeling standpoint.

13. Khan et al. (reference 14) previously reported SAXS analysis of several forms of the J6 E2 protein, but no comparison with those results was seen in the present study. Their SAXS results should be noted, and compared with the current SAXS results.

14. It is not clear why the authors selected PDB code 6MEI rather than 6MEJ, as the latter structure includes more residues in HVR1 (starting at residue 405 rather than residue 410) and has slightly higher resolution (2.8 Angstroms). A note on the selection of this structure should be included in the Results or Methods.

Reviewer #2: The authors construct three different E2 ectodomain structures and use molecular dynamics (MD) simulations to compare their dynamic properties. The overlap in the behavior of these ectodomains generally support their findings. As HVR2 is not hypervariable in genotype 2 it would of interest to extend the analysis of HVR2 by comparing J6 with the two genotype 1 isolates to see whether differences can be identified.

The authors apply a careful and methodical MD approach to increase our very limited understanding of HCV envelope protein E2 structure and function. The work appears to adhere to current MD standards and is performed in a clear fashion. It is nice that the authors provide SAXS evidence in support of their MD work. In addition, the discussion is relevant and well-balanced with a few exceptions listed below.

1) The potential contribution from missing HCV envelope proteins elements, most notably E1, seems skewed. Specifically, while the authors cite studies that suggest that HVR1 is on the surface of virions that does not, by itself, prove that it is not found there in an interaction with another envelope protein motif. Thus, if the interaction partner (e.g. E1) is missing then It does not seem overly strange that regions of E2, not constrained by adjacent secondary structure elements in E2 would be exhibiting high flexibility. While the authors interpretation of what their study means in terms of a disordered HVR1 is an intriguing one it seems the evidence for this is still lacking.

2) While the theory of a disordered HVR1 becoming ordered upon SR-BI engagement is intriguing it lacks support in the submitted manuscript. Partly due to the point made in bullet 1 on the reasons for HVR1 flexibility and partly because it assumes an HVR1-SR-BI interaction. However, HVR1-deleted sE2 can interact with SR-BI (Scarselli et al., 2002) if the right mutations are introduced and HVR1-deleted HCVcc can still depend on SR-BI during entry. This indicates that HVR1 is not the sole SR-BI interacting partner of E2 and perhaps suggests that other, indirect, effects of HVR1 deletion on SR-BI interaction are at play. Figure 9 seems particularly overreaching and it may even be considered misleading as the detailed atomic structures used to depict the hypothesis are not being used to establish the point being made (one would expect that atomic structures would be used to inform how and where the interaction occurs which is not the case). This is not to say that the findings regarding HVR1 movement correlating with changes throughout the E2 ectodomain (Figure 7) are not both compelling and extremely interesting, but rather it is a call to caution as to which interactions may precipitate this when considering full E1/E2 complexes on infectious particles interacting with a cell during entry.

Reviewer #3: The manuscript by Stejskal et al. reports an analysis of the structural and dynamic properties of the E” protein of the HCV virus. This protein is a key factor for HCV pathogenicity and is an important target for neutralizing MAbs. Characterizations of the E2 are also believed to hold important implications for the development of anti-HCV vaccines. Recent experimental investigations have provided an insightful but fragmentary picture of the protein structures. The main aim of this study is to complement available structural data with information derived by molecular dynamic simulations. Although MD studies on this system have been recently reported, the present study represent a significant advance in terms of simulation timescale, completeness of E2 structural model and covering of the strain variability of HCV. The work is technically sound and the data are clearly illustrated throughout the manuscript. The computational work is integrated by SAXS experiments that, although at low resolution, are in line with the MD results. On the basis of these considerations, I believe that the manuscript can be considered for publication. There are, however, some points that should be addressed:

1) The authors observed that E2 glycosylation has limited effects on the protein structure/dynamics. It is known, however, that E2 glycosylation, at least at specific sites, is essential for virus entrance and infectivity. The authors should comment this point.

2) The authors correlated the observed flexibility of the different E2 regions with their sequence variability detected in CV strains. In this framework, how do the authors explain the rather strict sequence conservation of the AS412 epitope with its remarkable flexibility (Fig. 2) (see also Balasco et al Curr Med Chem. 2017 24(36):4081-4101 and Int J Biol Macromol. 2018, 116:620-632 for a discussion on this feature that is also observed in other HCV proteins).

3) The authors performed a re-refinement of the PDB models before starting the MD. They found an improvement of the value of the crystallographic indicator R-factor (what about the R-free?). In my opinion this is very good a practice. The authors found a pattern of cis/trans peptide bonds that is closer to that observed in more recent E2 structures. Does this also reflect in a pair-wise analysis of the RMSD values of the core of the protein among these different E2 structures? I wonder whether the deposited models within the PDB should be amended to consider these improvements. Did the authors also try in their re-refinement process the RE_DO automated server?

**Have all data underlying the figures and results presented in the manuscript been provided?**

Reviewer #1: None

Reviewer #2: Yes

Reviewer #3: Yes

PLOS authors have the option to publish the peer review history of their article (what does this mean?). If published, this will include your full peer review and any attached files.

Reviewer #1: No

Reviewer #2: No

Reviewer #3: No

---

## [Decision Letter · Decision Letter 1]

29 Jan 2020

Dear Dr. Grove,

Thank you very much for submitting your manuscript "Flexibility and intrinsic disorder are conserved features of hepatitis C virus E2 glycoprotein." for consideration at PLOS Computational Biology. As with all papers reviewed by the journal, your manuscript was reviewed by members of the editorial board and by several independent reviewers. The reviewers appreciated the attention to an important topic. Based on the reviews, we are likely to accept this manuscript for publication, providing that you modify the manuscript according to the review recommendations.

Sincerely,

Guanghong Wei

Associate Editor

PLOS Computational Biology

Rob De Boer

Deputy Editor

PLOS Computational Biology

[LINK]

Reviewer's Responses to Questions

**Comments to the Authors:**

Reviewer #1: All comments have been addressed by the authors.

One minor comment is that the authors don't seem to specify the type of RMSD (backbone, C-alpha, or all-atom) used at several points, including the new Table S1 and the legend for Figure 5. The authors do, at one point, note that backbone atoms are used for RMSF calculations for the MD output in the Methods, so it is likely that they also use backbone atoms for RMSD calculations between structures. However, for clarity and to avoid ambiguity, the authors should specify in Table S1, Figure 5, and elsewhere as needed that backbone RMSDs are being used, or another type of RMSDs as warranted.

Reviewer #2: The authors construct three different E2 ectodomain structures and use molecular dynamics (MD) simulations to compare their dynamic properties. The overlap in the behavior of these ectodomains generally support their findings. As HVR2 is not hypervariable in genotype 2 it would of interest to extend the analysis of HVR2 by comparing J6 with the two genotype 1 isolates to see whether differences can be identified.

Author response:

We analysed alignments of GT1 and GT2 HVR-2 for sequence diversity and observed equivalent sequence divergence (25-30%) in both genotypes. Therefore, we do not believe further focus on HVR-2 is appropriate. The HVR-2 alignments used for this analysis can be provided if necessary.

Reviewer response:

This is likely because the authors did not stratify their analysis to subtypes. In fact, HVR2 of gt 2a is quite different from HVR2 of gt 2b, but they are both quite conserved within-subtype. Similarly, gt 1a is fairly conserved, whereas gt 1b has a high degree of sequence diversity, particularly in the C-terminal part of HVR2. I think these differences are of interest and warrant further investigation, but I do not consider it critical for the current manuscript.

The authors apply a careful and methodical MD approach to increase our very limited understanding of HCV envelope protein E2 structure and function. The work appears to adhere to current MD standards and is performed in a clear fashion. It is nice that the authors provide SAXS evidence in support of their MD

work. In addition, the discussion is relevant and well-balanced with a few exceptions listed.

1. The potential contribution from missing HCV envelope proteins elements, most notably E1, seems skewed. Specifically, while the authors cite studies that suggest that HVR1 is on the surface of virions that does not,

by itself, prove that it is not found there in an interaction with another envelope protein motif. Thus, if the interaction partner (e.g. E1) is missing, then It does not seem overly strange that regions of E2, not constrained

by adjacent secondary structure elements in E2 would be exhibiting high flexibility. While the authors interpretation of what their study means in terms of a disordered HVR1 is an intriguing one it seems the evidence for this is still lacking.

Author response:

We broadly agree with the reviewer, but we would argue that we included a clear caveat to our work that discusses the limitations of using monomeric E2 and integrates the best current evidence for which regions are likely to buried in the E1E2 complex. Nonetheless we have added a further statement on line 632 to reassert this important limitation.

Reviewer response:

Firstly, there is no added statement at line 632, which is in the Methods section of the paper. I assume that the lines referred to 600-601? Secondly, the adjustment line added is missing the point I was raising. While I do appreciate the caveat present in the initial submission it is not simply a matter of a region being either buried or exposed on E1E2 (on particles) or soluble E2, respectively. Rather, a region can be at the surface of a protein without being highly flexible. Thus, flexibility of HVR1 and AS412 on soluble E2 are not necessarily evidence of flexibility in higher order structures in the context of E1E2. This is the point I think it would be good to make explicitly.

2. While the theory of a disordered HVR1 becoming ordered upon SR-BI engagement is intriguing it lacks support in the submitted manuscript. Partly due to the point made in bullet 1 on the reasons for HVR1 flexibility

and partly because it assumes an HVR1-SR-BI interaction. However, HVR1-deleted sE2 can interact with SR-BI (Scarselli et al., 2002) if the right mutations are introduced and HVR1- deleted HCVcc can still depend

on SR-BI during entry. This indicates that HVR1 is not the sole SR-BI interacting partner of E2 and perhaps suggests that other, indirect, effects of HVR1 deletion on SR-BI interaction are at play. Figure 9 seems

particularly overreaching and it may even be considered misleading as the detailed atomic structures used to depict the hypothesis are not being used to establish the point being made (one would expect that atomic

structures would be used to inform how and where the interaction occurs which is not the case). This is not to say that the findings regarding HVR1 movement correlating with changes throughout the E2 ectodomain

(Figure 7) are not both compelling and extremely interesting, but rather it is a call to caution as to which interactions may precipitate this when considering full E1/E2 complexes on infectious particles interacting with

a cell during entry.

Author response:

On the matter of sE2 interacting with SR-B1 via HVR-1: the reviewer is referring to Fig. 3C of Scarselli et. al. in which delta HVR-1 sE2 with adaptive mutations (V514M and L615H) exhibit enhanced binding to HepG2 cells. In this particular experiment the binding of sE2 is likely to be multimodal (potentially also binding to attachment factors, such as HSPG), therefore it is not a good system for specifically examining E2-SR-B1 interactions. The cleanest system to study this is CHO cells expressing SR-B1; Scarselli et. al. use these cells to demonstrate

complete loss of SR-B1 binding upon deletion of HVR-1 (Fig 8 of that paper). This observation has been recapitulated by two other group (http://www.jbc.org/content/287/42/35631.long,

http://www.jbc.org/content/287/37/31242.long). Therefore, we assert that the best evidence suggests that E2-SR-B1 interactions are HVR-1 dependent. We accept that the proposed significance of E2-SR-B1 interaction is quite

speculative and understand the reviewer’s concerns; therefore, we have reworded the text to soften this proposal (line 600) and have moved the associated figure to the supplementary material.

Reviewer response:

Firstly, line 600 is not the place where adjustments were made. Are the authors referring to line 575-580? Also, there appears to be a comment associated with this section. Secondly, I fully agree that evidence exists for an SR-BI interaction with HVR1. However, evidence is also abundant that it is much more complicated than a “simple” interaction (for which the two papers highlighted in the author response are good examples) and could involve many other “partners” both within and outside the envelope proteins themselves. In the authors response they write that the interaction is “HVR1-dependent”, which is very different from writing that the interactions occur via HVR1 as they do in the Introduction. Both could be true and I do not have any issue with the speculations on what the significance of the MD simulations could be in terms of potential SR-BI interactions with HVR1, as long as these limitations are clear. This is particularly important when the authors had a figure in the main article showing this interaction using high resolution molecular structures in the absence of these important caveats. Moving it to the supplementary was a good idea.

Reviewer #3: The authors properly addressed the issues I raised.

In some places crystal is misspelled as crytsal

**Have all data underlying the figures and results presented in the manuscript been provided?**

Reviewer #1: Yes

Reviewer #2: Yes

Reviewer #3: Yes

PLOS authors have the option to publish the peer review history of their article (what does this mean?). If published, this will include your full peer review and any attached files.

Reviewer #1: No

Reviewer #2: No

Reviewer #3: No
---

## [Editor Report · Decision Letter 2]

4 Feb 2020

Dear Dr. Grove,

We are pleased to inform you that your manuscript 'Flexibility and intrinsic disorder are conserved features of hepatitis C virus E2 glycoprotein.' has been provisionally accepted for publication in PLOS Computational Biology.

Before your manuscript can be formally accepted you will need to complete some formatting changes, which you will receive in a follow up email. A member of our team will be in touch within two working days with a set of requests.

Best regards,

Guanghong Wei

Associate Editor

PLOS Computational Biology

Rob De Boer

Deputy Editor

PLOS Computational Biology

---

## [Editor Report · Acceptance letter]

24 Feb 2020

PCOMPBIOL-D-19-01295R2 

Flexibility and intrinsic disorder are conserved features of hepatitis C virus E2 glycoprotein.

Dear Dr Grove,

I am pleased to inform you that your manuscript has been formally accepted for publication in PLOS Computational Biology. Your manuscript is now with our production department and you will be notified of the publication date in due course.

With kind regards,

Sarah Hammond
